

# Future climatic drivers and their effect on $PM_{10}$ components in Europe and the Mediterranean Sea

Arineh Cholakian[1,2], Augustin Colette[2], Giancarlo Ciarelli[1], Isabelle Coll[1], and Matthias Beekmann[1]

[1]Laboratoire Inter-Universitaire des Systèmes Atmosphériques (LISA), UMR CNRS 7583, Université Paris Est Créteil et Université Paris Diderot, Institut Pierre Simon Laplace, Créteil, France
[2]Institut National de l'Environnement Industriel et des Risques, Parc Technologique ALATA, Verneuil-en-Halatte, France

**Correspondence:** Arineh Cholakian (arineh.cholakian@lisa.u-pec.fr)

**Abstract.** Multiple CMIP5 future scenarios are compared to historic simulations in order to study different drivers governing air pollution: Regional climate, anthropogenic emissions and long-range transport. Climate impact study covers the period of 2031 to 2100 for future scenarios compared to 1976 to 2005 for historic simulations, and includes three RCPs (Representative concentration pathways ,RCP2.6, RCP4.5 and RCP8.5). A detailed analysis of total $PM_{10}$ concentrations, its changes and also that of its components is included. The individual effects of meteorological conditions on $PM_{10}$ components are explored in these scenarios in an effort to pinpoint the meteorological parameter(s) governing each component. Anthropogenic emission impact study covers the period of 2046 and 2055 with CLE2050 (Current legislation emissions for 2050) anthropogenic emissions compared to CLE2010 in historic simulations covering the period of 1996 to 2005. Long-range transport is explored by changing the initial and boundary conditions in the chemistry-transport model, these scenarios cover the same period as the emission impact studies. Finally, a cumulative effects of these drivers is performed and the contribution of each driver on $PM_{10}$ and its components is calculated. The results show that, regional climate causes a decrease in $PM_{10}$ concentration in our scenarios, as a result of a decrease in nitrate, sulfate, ammonium and dust in most scenarios. Meanwhile, biogenic secondary organic aerosols (BSOA) shows an important increase in all scenarios. Nitrate and BSOA show a strong dependence to temperature, while sulfates are dependent to relative humidity. A cumulative look at all drivers shows that anthropogenic emission changes overshadow changes caused by climate and long-range transport for most components except for dust, for which long-range transport changes seem to be more influential.

## 1 Introduction

Particulate matter (PM) is one of the most important constituents of air pollution. It can have a variety of adverse effects on air quality (Seinfeld and Pandis, 2016) and subsequently on human health ( Pope and Dockery, 2006 ; Kampa and Castanas, 2008 ; Anderson et al., 2012; Im et al., 2018) and ecosystems (Grantz et al., 2003). Studies have shown that the life expectancy of the population can change drastically in areas densely polluted by atmospheric aerosols (Pope et al., 2009). PM comprise a large number of different components, each one having different origins and very diverse behaviors with respect to meteorological parameters. Therefore, there are many different ways in which the particles can affect air quality, making their investigation both important and complex. The intricacy of studying PM becomes deeper when coupling its effects with climate change




since air quality and climate change have intertwined interactions (e.g. Kinney, 2008; Wild, 2009; Seinfeld and Pandis, 2016). In other words, changes in meteorological conditions have varied effects on air quality, but at the same time climate change may be affected by radiative forcing of air pollutants. These effects can, in some cases, go in the same direction as each other, or they may show opposite outcomes. Thus, when exploring the future air quality, it is important to consider different drivers of air quality and, if possible, to examine the effects of each driver separately.

Air pollution is governed mainly by four factors; anthropogenic and/or biogenic emissions of primary pollutants and precursors of secondary pollutants, atmospheric chemistry, long-range transport and of course, meteorology (Jacob and Winner, 2009). While these factors are listed separately, they have interactions among themselves. For example, atmospheric chemistry is directly affected by temperature and radiative forcing. Similarly, parameters such as precipitation, wind speed and wind direction can have positive or negative effects on dispersion and deposition. Also, indirectly, meteorological conditions such

as temperature and wind speed can have impacts on the emission of primary pollutants, which may also be precursors of secondary pollutant (EEA, 2004). As a result, the sensitivity of air quality to climate change seems to be crucial, but complex to investigate.

The sensitivity to climate change of different areas in the world depends on the existing local meteorological conditions. Giorgi, (2006) calculated a factor to determine the climate change hotspots in future scenarios to show the sensitivity of

different regions when faced with climate change. Using the differences between wet season and dry season historic and future precipitation and temperature for different regions and an ensemble of scenarios and models, he has shown that the Mediterranean and the north-eastern European area are more sensitive to climate change than the other regions of the world. According to his calculations, European area and the Mediterranean as a whole, are one of the most important hotspots for climate change. This highlights the importance of understanding the changes that might affect these regions. Therefore, the

focus of this study will be on the European area with a special attention to the Mediterranean basin. This is why the current work is related to the ChArMEx (the Chemistry Aerosol Mediterranean Experiment; http://charmex.lsce.ipsl.fr) project. The goals of ChArMEx are to better assess the sources, formation, transformation and mechanisms of transportation of gases and aerosols in the western Mediterranean basin and also to better estimate the future composition of the atmosphere of the Mediterranean Sea. The measurement part of this campaign took place in the western Mediterranean basin during the period of 2012-2014,

the analysis of the data obtained during the campaign and the assessment of future atmospheric changes for the basin are still ongoing.

A regional Chemistry-Transport Model (CTM) was used to explore future possible changes in these regions. Running such regional simulations requires inputs from a global CTM, a Global Circulation Model (GCM) and a Regional Climate Model (RCM), as well as anthropogenic/biogenic emission inputs. Changes made to these inputs make it possible to distinguish the

effects of different drivers on air pollution one by one. Modifying RCM inputs allows the estimation of the effects of meteorology alone, while a combined modification of RCM and global CTM inputs allows us to simultaneously assess the impacts of meteorology and long-range transport. On the other hand, apart from RCM inputs, changes in anthropogenic emissions allow to explore the effects of meteorology and emissions on air pollution.





These kinds of studies of course already exist for different parts of the world, for one or multiple drivers and for different
components. For example, Liao et al., 2006 used a global model to explore the atmospheric changes expected in the year 2100,
by comparing a year of historic simulations to a yearlong simulation in 2100, where all factors were changed. The first study
that investigated the future atmospheric conditions of European area focused only on ozone changes and used the two 30-year
long future scenarios compared to a 30 year-long historic period (Meleux et al., 2007). Other studies use an ensemble of future
simulations, each with a different model, in order to compare the results given by each of these models (e.g. Langner et al.,
2012). Based on the IMPACT2C project (Jacob, 2017), Lacressonnière et al., 2016 and Lacressonnière et al., 2017 focus on
European regional simulations, exploring the effects of a 2°C climate change combined with anthropogenic emission changes
in an ensemble of four models. Similarly, Fortems-Cheiney et al., 2017 explored the same scenarios for a 3°C of climate
change, with a focus on gaseous species, and Carvalho et al., 2010 conducted a SRES A2 climate change scenario over the end
of the 21st century for Europe, zooming on Portugal. The common point of all these studies is that all the impacting factors
have been changed simultaneously in a future scenario. A review of existing scenarios is presented in Colette et al. (2015).

Unlike these studies, other papers have investigated the impact of emissions and meteorology on atmospheric composition in
future scenarios separately. Dawson et al., 2007 focused on determining the atmospheric sensitivity to changes in meteorolog-
ical conditions in the Eastern US, over a simulation period of two months. Megaritis et al., 2014 have used a similar approach
as Dawson et al. (2007), exploring the sensitivity of the atmospheric composition to changes in meteorological conditions for
Europe over a three month-long simulation period. These two studies have conducted sensitivity tests on short (months-long)
periods of time. Lemaire et al., (2016) have explored climate change effects from the same dataset as the one used in our work,
developing a statistical method to ascertain the meteorological parameters that affects atmospheric pollutants in future scenar-
ios. Hedegaard et al., 2013 also looked at the relative importance of emissions and meteorological drivers in a hemispheric
model. Finally, Colette et al. (2013) explored the same scenarios that we worked on with the aim of analyzing the global effects
of the three drivers (meteorology, emissions and boundary conditions) on atmospheric composition, although only focusing on
Europe as a whole and not investigating the individual effect of the drivers on PM composition. Our aim is to complement the
previous works by providing a deeper insight into the respective impacts of climate, atmospheric content and emission-related
forcing. This is why the work described here focused on simulating a set of future and climatic scenarios over long periods of
times, and observing the differences between the drivers discussed above. The chosen approach is to change the drivers one by
one and to assess the differences induced in PM components, in order to access the individual effects of the parameters of the
simulation.

It should be noted that other studies have explored the dependence of PM components on meteorological conditions as well
(Dawson et al., 2007 ; Carvalho et al., 2010 ; Fiore et al., 2012 ; Jiménez-Guerrero et al., 2012 ; Juda-Rezler et al., 2012 ;
Hedegaard et al., 2013 ; Megaritis et al., 2014). Still, most of these studies were performed for short time-periods, such as
1 year-long simulations (shorter in most cases) and several used sensitivity tests and not actual future scenarios to assess the
changes of different meteorological parameters. Conversely, Lemaire et al., (2016) have explored the sensitivity of ozone and
$PM_{2.5}$ to different meteorological parameters, using 30 years of RCP8.5 scenario simulations. However, they do not consider
in detail the relationship between the speciation of PM components and these parameters. To the best of our knowledge, the




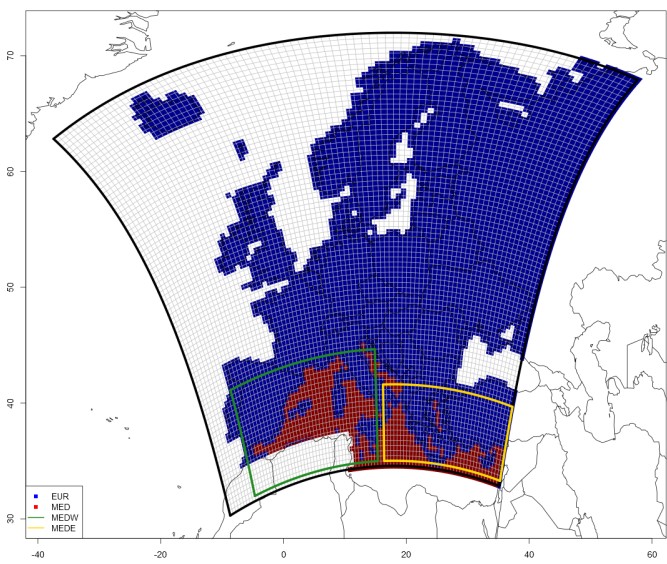

**Figure 1.** Extension of the main domain and sub-domains. Four sub-domains are used in this study : EUR – containing only continental Europe (blue cells), MED – containing only the Mediterranean Sea (red cells), MEDW – western Mediterranean region (green regtangle), MEDE – eastern Mediterranean region (yellow rectangle).

sensitivity of PM components to meteorological parameters for a dataset this extensive containing multiple scenarios and calculation of effects of different drivers on same data pool has not been done up to now.

In this paper, after a brief introduction to the simulations and the modeling framework, the impacts of different drivers will be explored. The analysis first deals with climate impacts; then, the effects of long-range transport and emission changes will be discussed. Finally, the impact of each of these three drivers on the concentration of $PM_{10}$ and its components will be calculated. The discussions of the results will be divided into two parts corresponding to geographic areas: the European and the Mediterranean sub-domains. Finally, a prospective view of what the PM component concentrations in the Mediterranean basin may be like at the end of the 21st century will be given in the last part of the paper.

## 2  Method

In this section, we introduce the architecture of the modelling framework, with a focus on the most sensitive component which is the chemistry transport model. We also provide references to the input data used by the CHIMERE model in terms of future scenarios and for the various combinations of input parameters.



| | # | Simulation name | Simulation period | Global chemistry model | Regional climate model | Anthropogenic emissions |
|---|---|---|---|---|---|---|
| Climate | 1 | Hist | 1976-2005 | LMDZ-INCA-RCP2.6 | WRF Historic | ECLIPSE-V4a CLE2010 |
| | 2 | RCP2.6 | 2031-2100 | | WRF RCP2.6 | |
| | 3 | RCP4.5 | 2031-2100 | | WRF RCP4.5 | |
| | 4 | RCP8.5 | 2031-2100 | | WRF RCP8.5 | |
| BC | 5 | RCP4.5-BC | 2046-2055 | LMDZ-INCA-RCP4.5 | WRF RCP4.5 | ECLIPSE-V4a CLE2010 |
| Emissions | 6 | RCP4.5-CLE2050 | 2046-2055 | LMDZ-INCA-RCP2.6 | WRF RCP4.5 | ECLIPSE-V4a CLE2050 |
| | 7 | RCP4.5-MFR2050 | 2046-2055 | | | ECLIPSE-V4a MFR2050 |
| all | 8 | RCP4.5-BC-CLE2050 | 2046-2055 | LMDZ-INCA-RCP4.5 | WRF RCP4.5 | ECLIPSE-V4a CLE2050 |

**Table 1.** The different scenarios used in this work

## 2.1 Modeling framework

The assessment of the long term evolution of air quality in the context of a changing climate is performed with a suite of deterministic models following the framework introduced by Jacob and Winner (2009). Global climate projections are obtained from a global circulation model (GCM) that feeds a global chemistry-transport model and a regional climate model. Finally, the latter two drive a regional chemistry transport model. The setup used in this study is presented in detail in Colette et al., (2013) and Colette et al., (2015).

The global circulation model is the IPSL-CM5A-MR large-scale atmosphere-ocean model (Dufresne et al., 2013). It provides input to the regional climate model and the global chemistry-transport model with global meteorological fields. It uses LMDz (Hourdin et al., 2006) as its meteorological model, ORCHIDEE (Krinner et al., 2005) as its land surface model, NEMO (Madec and Delecluse, 1998) and LIM (Fichefet and Maqueda, 1999) being respectively the oceanic and the sea-ice models. The horizontal resolution of this global model is 2.5°×1.25° with 39 vertical levels. For each scenario, the corresponding RCP

is used for the anthropogenic radiative forcing. The Weather Research and Forecasting model (WRF, Wang et al., 2015) is used as the Regional Climate Model (RCM). The regional climate simulations were part of the EURO-CORDEX (Jacob et al., 2014) with a spatial resolution of 0.44°. The historic simulations were evaluated by comparison to experimental data (Menut



et al., 2012 ; Kotlarski et al., 2014 ; Katragkou et al., 2015). The LMDZ-INCA (Hauglustaine et al., 2014) global CTM is used for the production of chemical initial/boundary conditions for the regional CTM.

## 2.2 CHIMERE CTM

The CHIMÈRE offline regional CTM has been widely used both for future scenarios (Colette et al., 2015 ; Lacressonnière et al., 2016) and for research activities in France (Zhang et al., 2013 ; Petetin et al., 2014 ; Menut et al., 2015 ; Rea et al., 2015 ; Cholakian et al., 2018) and abroad (Hodzic and Jimenez, 2011). In this work, the 2013b version of the model has been used for all simulations (Menut et al., 2013). The simulations were conducted with the Euro-CORDEX domain with a horizontal resolution of 0.44° and 9 vertical levels ranging from the surface to 500mb. The aerosol module is ran with a simple two-product scheme for the simulation of Secondary Organic Aerosols (SOA, Bessagnet et al., 2008) and with the ISORROPIA module for the simulation of inorganic aerosols (Fountoukis and Nenes, 2007). It provides simulated aerosol fields including EC, sulfate, nitrate, ammonium, SOA, dust, salt and PPM (primary particulate matter other than the ones mentioned above) considering coagulation, nucleation and condensation processes, as well as dry and wet deposition. The land-use data come from Globcover (Arino et al., 2008) with a base resolution of $300{\times}300m^2$ which does not change in different series of simulations, therefore. Therefore, land-use changes are not discussed in the current work.

The simulation domain has a 0.44° resolution (figure 1). The analysis is performed using the sub-domains presented in figure 1. The EUR sub-domain only concerns the European continent (including British Isles) and a land-sea mask was used to remove other parts of the domain. The MED sub-domain was produced using a land-sea mask as well, but this time it only contains the Mediterranean Sea. The MEDW and MEDE are the last two sub-domains. They refer to the western and eastern Mediterranean areas respectively. It is important to bear in mind that these two sub-domains, contrary to the previous ones, contain both land and sea for the purpose of seeing the effects of enclosing lands on the Mediterranean area. Due to this set-up, the sum of MEDW and MEDE is different from MED.

## 2.3 Climate scenarios

Representative Concentrations Pathway scenarios designed for the fifth IPCC report (Meinshausen et al., 2011; van Vuuren et al., 2011b) are used. Simulations with three of these CMIP5 RCPs (Taylor et al., 2012; Young et al., 2013) are selected: RCP2.6, RCP4.5 and RCP8.5 respectively consider 2.6 W.m$^{-2}$, 4.5 W.m$^{-2}$ and 8.5 W.m$^{-2}$ of radiative forcing at the end of the 21st century. It is worth noting that RCP8.5 includes by far the least mitigation policies compared to the other two scenarios, therefore resulting in a high radiative forcing at the end of the century, with a temperature increase comprised between 2.6 and 4.8°C for Europe according to EEA (European Environmental Agency ). On the contrary, RCP2.6 scenario considers a radiative forcing leading to a low range temperature increase by 2100 (between 0.3 and 1.7°C). This means, it has to consider ambitious greenhouse gas emissions reductions as well as carbon capture and storage. The RCP4.5 is an intermediate scenario with less stringent climate mitigation policies, which results in a temperature increase range comprised between the two extreme scenarios mentioned before.



## 2.4 Air pollutant Emissions

The biogenic emissions input is taken from the Model of Emissions of Gases and Aerosols from Nature, v2.04 (MEGAN, Guenther et al., 2006), it is worth mentioning that the emission factors and the Leaf Area Indexes (LAI) provided by this model are the same for all simulations, but since many of the biogenic gases have a temperature-dependent nature, their emissions increase the higher the temperature goes. The MEGAN version used in CHIMERE takes into account the dependence of emissions from six BCOVs (isoprene, $\alpha$-pinene, $\beta$-pinene, humulene, limonene and ocimene) to temperature, radiation and LAI.

Anthropogenic emission are taken from the global emissions projections ECLIPSE-V4a (Amann et al., 2013; Klimont et al., 2013, 2017). It covers the time period 2005-2050 with two main pathways for either the Current Legislation Emissions (CLE) or Maximum Feasible Reduction (MFR) which show the effects of minimum and maximum of mitigation effort that can happen until 2050, therefore giving us a spectrum of possible influences of anthropogenic emissions in future scenarios. For both scenarios, the atmospheric emission of the main pollutants are available as global maps at 0.5° resolution.

## 2.5 Simulations

The simulations used for this study are shown in table 1. The goal of this study is to separately look at different drivers used in the regional CTM (Chemistry-Transport Model) that can affect the results of future simulations. Therefore, different series of simulations were performed for evaluating the climate impacts, emission impacts and boundary condition impacts on PM concentrations. Changing each of these factors means changing regional climate, anthropogenic emissions and long-range transport respectively.

For the climate impact study, a range of simulations are used with RCP2.6, RCP4.5 and RCP8.5 scenarios using constant anthropogenic emissions and boundary conditions. These scenarios will be compared to historic simulations.

To explore changes induced by boundary conditions, RCP4.5 scenarios with two different sets of boundary conditions will be compared to historic simulations. In order to explore the effects of changes in boundary conditions, two versions of inputs from the same global CTM are used. The difference between these two is the fact that for one of them anthropogenic pollutants from the RCP emissions (emissions produced for RCP scenarios exclusively) are used as anthropogenic inputs while for the other one relies on ECLIPSE-V4a emissions are used as these inputs, the former is presented in detail in Szopa et al., 2006, 2013.

The emission impact study uses the RCP4.5 climate forcing with CLE 2010, CLE 2050 and MFR 2050 anthropogenic emissions, each will be again compared to historic simulations.

As seen in table 1, the period for different series of simulations is different, for climate impact studies a 30 year long historical simulation and 70 year long periods for each future scenario will be used, for the rest of the simulations 10 years of historical simulations (1996 to 2005) and 10 years of future scenarios representing the 2050s (2045 to 2054) will be used. This latter period was chosen because it is centered on year 2050 for which CLE/MFR 2050 emissions are available. At the end, a ten year long simulation is performed where all these factors change simultaneously, which should give a more realistic





view of what the climate of 2050s would be if certain hypothesis were made. It would have been numerically too strenuous to perform 70 years of simulations for emission/boundary condition drivers bearing in mind that the climatic impact simulations
already mount up to 240 years of simulations.

A validation of the historic period simulations was done for the meteorological parameters in Menut et al., (2012) and Kotlarski et al., (2014), while a validation of some chemical species is presented in this article in the supplementary materials (SI-1) using an annual profile of 10 years of historic simulations between 1996 to 2005 (with 2010 constant anthropogenic emissions) compared to an annual profile of all available measurements in EEA and airbase stations between 2005 to 2015
(EEA, 2016).

## 3    Climate impacts

This section discusses the comparisons between the simulations presented as RCP2.6, RCP4.5 and RCP8.5 with historic simulations (simulations 1 to 4 in table 1). Since all the inputs except the meteorological meteorological fields remain the same in these 4 series of simulations, it is possible to disentangle the effects of climate alone on PM concentration in different RCPs.
While the specific goal of the paper is to focus on only the PM changes in the Mediterranean area, a general overview at the European domain is also provided. Before exploring PM changes, temperature changes will be analyzed as they imply important effects on biogenic VOC emissions which are precursors for the production of secondary organic aerosols (SOA) and on gas/particle phase partitioning in simulations. The dependency of total $PM_{10}$ and its components to other meteorological parameters will also be discussed in detail in this section.

### 3.1    Meteorological parameters

The RCP2.6 reaches its maximum radiative forcing (2.6 W.m$^{-2}$) in the 2040s (van Vuuren et al., 2011a), meaning that an increase in radiative forcing is seen in this scenario until this year, while afterwards we observe a continuous decrease. Therefore, in the bulk of RCP2.6 simulations over Europe, a temperature decrease appears from about 2050 onwards, followed by a new increase in temperature over the last 10 years simulated (figure 2). As for the RCP4.5 scenario, this maximum radiative
forcing is reached in the 2070s (Thomson et al., 2011) while such a maximum is not reached for RCP8.5 until the end of the century. As a consequence, in RCP4.5 the temperature increase levels out after about 2070 while for RCP8.5 temperature keeps increasing over the whole period in RCP8.5. Those elements explain the larger similarities of RCP4.5 and RCP8.5 as well as their structural differences with RCP2.6. These time evolutions are similar for the European (EUR) and Mediterranean (MED) domains, albeit absolute temperatures are nearly 10 degrees warmer over the Mediterranean in average (SI2 fig.1 for
2D temperature fields).

In the following comparisons, we calculated annual average values over each sub-domain and for each parameter and time series corresponding to 30 years of historic simulations and 70 years of future scenarios are presented. The uncertainties associated with each value account for the spatial variability of each parameter.



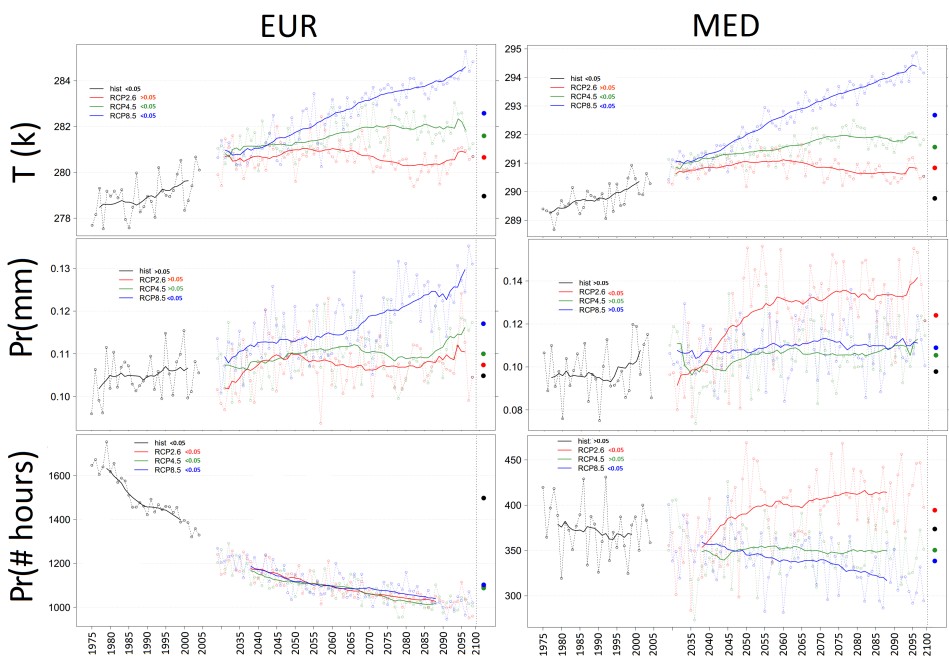

**Figure 2.** Time series of temperature (K), precipitation (mm) and number of rainy hours (from the top to bottom) for EUR and MED sub-domains (from left to right) for all climate change scenarios as well as historic simulations. The average for each scenario is shown by the points at right side of the plot. The solid lines show the rolling average of 30 years for future scenarios and 20 years for historic scenarios. Numbers in the legend show the p-value of the linear regression for each scenario.

Two-meter Temperatures are generally higher for the Mediterranean area than for the European continent, however, the
increase in temperature is more pronounced for EUR than for MED (+1.69, +2.63 and +3.62° for EUR and +1.06, +1.79
and +2.91° for MED for RCP2.6, RCP4.5 and RCP8.5 respectively). In the two other Mediterranean sub-domains temperature
changes are higher for MEDE than for MEDW (+0.73, +1.77 and +2.88° for MEDW and +1.25, +2.05 and +3.38° for MEDE for
RCP2.6, RCP4.5 and RCP8.5 respectively). While the European sub-domain shows a more important temperature increase in
winter (+1.86, +3.44 and +4.42° for RCP2.6, RCP4.5 and RCP8.5 respectively), the Mediterranean sub-domains show a larger
increase in summer (for MED +1.40, +2.06 and +3.22° for RCP2.6, RCP4.5 and RCP8.5 respectively). Maps of differences in
temperature for the different scenarios and seasons are presented in the supplementary materials (SI2 fig.2). It should be noted
that the average temperature change discussed here agree with the literature (Knutti and Sedlàçek, 2012; Vautard et al., 2014).

As seen also in figure 2, some of the parameters behave differently in the two sub-domains. For example, for the precipitation
amount, an increase is simulated in EUR for RCP8.5 while a slight decrease is simulated for RCP2.6 after the 2050s, which
makes precipitation stronger on the average in RCP8.5 than in RCP2.6 for the future period. In the MED area, a rather opposite
behavior is noted: precipitation is stronger in RCP2.6 than in RCP8.5. On the contrary, there is a steady decrease in the number
of rain episodes (sum of hours rained each year in each scenario, a threshold being fixed for each sub-domain using the average
of 25th percentile for the whole duration of simulations). Therefore, rain events are expected to become more intense (Vautard





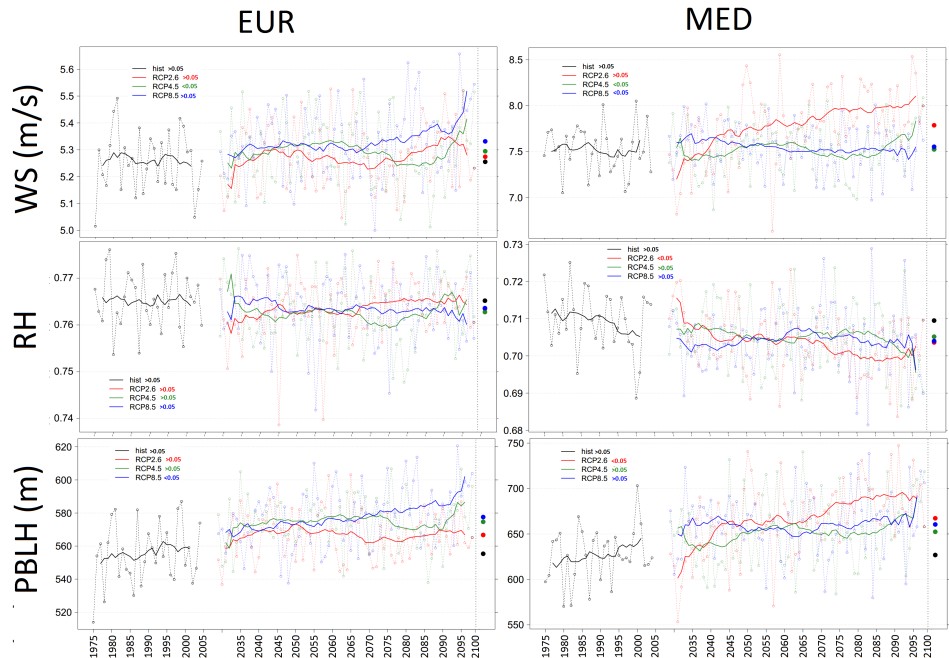

**Figure 3.** Time series of wind speed (m.s-1), relative humidity and PBL height (m) (from the top to the bottom) for EUR and MED sub-domains (from left to right) for all climate change scenarios as well as historic simulations. The average for each scenario is shown by the points at right side of the plots. The solid lines show the rolling average of 30 years for future scenarios and 20 years for historic scenarios. Numbers in the legend show the p-value of the linear regression for each scenario.

et al., 2014). As for the number of episodes, the same result as EUR is obtained, except for RCP2.6 where an increase in the number of rainy hours is simulated. This increase corresponds to the western basin of the Mediterranean Sea. For the Eastern basin, all scenarios show a decrease in this parameter (SI3 fig1 for MEDW and MEDE).

The same type of comparison is performed for wind speed (10m wind), relative humidity and PBL height in figure 3 for EUR and MED. 2D maps for these parameters are presented in SI (SI3 fig.2). Relative humidity remains rather constant without significant trend over EUR and MED, with the exception of RCP2.6 for MED showing a significant decrease. Wind speed shows mostly non-significant trends, with the exception of RCP2.6 over the MED domain. The results for wind speed changes were also seen in other studies (Dobrynin et al., 2012; de Winter et al., 2013). Finally, PBL height increases are significant for RCP8.5 over EUR and for RCP2.6 over MED.

These meteorological parameters have interactions between themselves. The values of correlations between different meteorological parameters examined in this work are shown in SI (SI4).



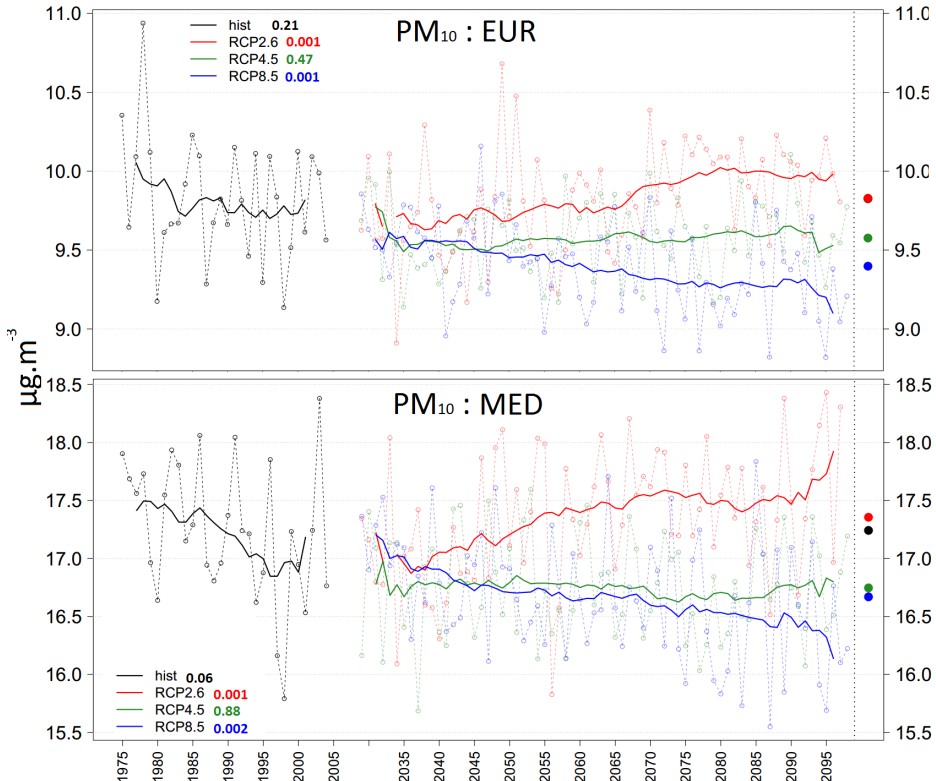

**Figure 4.** $PM_{10}$ time series for EUR and MED sub-domains for all climate change scenarios and historic simulations. The average for each scenario is shown by the points at right side of the plot. The solid lines show the rolling average of 30 years for future scenarios and 20 years for historic scenarios. Numbers in the legend indicate the p-value of the linear regression for each scenario.

## 3.2  $PM_{10}$ concentrations

The simulated $PM_{10}$ concentrations are shown in figure 4 for the EUR and MED sub-domains. The difference between future (2031 - 2100) and historical (1976 – 2005) simulations are presented, with historical simulations being used as a reference. Compared with historical simulations, we observe a decrease in $PM_{10}$ for all scenarios, for both EUR ($0\pm0.95\%$, $-2.57\pm0.90$ and $-4.40\pm0.87\%$ for RCP2.6, RCP4.5 and RCP8.5 respectively) and MED ($-1.77\pm0.09\%$, $-5.65\pm0.11\%$ and $-8.10\pm0.12\%$ for RCP2.6, RCP4.5 and RCP8.5 respectively). The uncertainties shown here and in the rest of the document refer to spatial one sigma intervals. The reasons for these changes in $PM_{10}$ will be discussed in the next sub-section, by analyzing individual PM components.

Alternatively, we also calculate linear trends for the future periods. A statistically significant positive trend is observed for RCP2.6 in the future, while it is significantly negative for RCP8.5, and non-significant for RCP4.5 (p-values are given in figure 4 for linear trend lines).





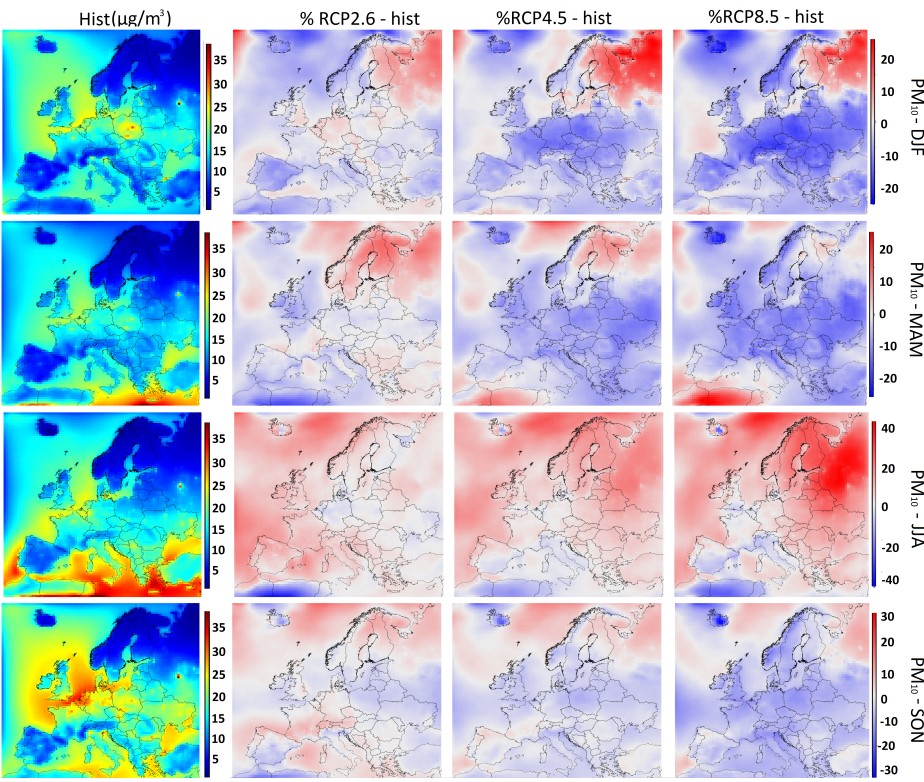

**Figure 5.** $PM_{10}$ seasonal average concentrations in the historical simulation (first column). Relative differences between the climate simulations (RCP2.6, RCP4.5 and RCP8.5) and the historical simulation (hist) (second, third and four columns) Rows represents the different seasons (winter, spring, autumn and summer, from top to the bottom).Please notice that a different scale is used between the seasons in order to facilitate the reading of the panel.

Evidently, $PM_{10}$ has a seasonal variation which is shown in figure 5, box plot graphs of $PM_{10}$ for all four seasons and all four sub-domains are shown in figure 6 (SI5 fig.1 for $PM_{2.5}$). Interestingly, for EUR, the general $PM_{10}$ decrease noted above for the scenarios HIST/RCP2.6/RCP4.5/RCP8.5 is reversed in the summer period. We can see that elevated concentrations of $PM_{10}$ are simulated over the Mediterranean area for all seasons, reaching their maximum in spring (figure 6). Another interesting result in figure 5 is the increasing concentrations of $PM_{10}$ over Eastern Europe in summer, and over the Scandinavian and Eastern Europe both in summer and winter, in RCP4.5 and RCP8.5. Yet, the same effect is not seen in RCP2.6 in any of these two seasons, highlighting therefore a structural difference between RCP2.6 and the other two scenarios. Such a result will be discussed in the light of $PM_{10}$ components and meteorological parameters covariance in section 3 4. A more general explanation for these structural differences can be found in the nature of the three scenarios, caused by the discriminated changes in meteorological parameters mentioned before (section 3 1).





### 3.3  Distribution of chemical $PM_{10}$ components

Figure 7 shows the distribution of $PM_{10}$ concentrations for all the sub-domains and for all climate impact simulations (SI5 fig.2 for $PM_{2.5}$). For each PM component, the relative differences between future scenarios and historical simulations are also reported. Major differences can be seen in the distribution of different PM components: for $PM_{10}$ the major contributors are salt and dust particles for the domains of our interest, while their contribution to $PM_{2.5}$ is lower. As a consequence, secondary inorganic ($SO_4^{2-}$, $NO_3^-$ and $NH_4^+$) and carbonaceous aerosol (BC, POA, BSOA) are the major contributors to $PM_{2.5}$ (SI5

fig.2). The most important changes in the future scenario outputs (with respect to historic simulations) are found to be a decrease in nitrates (for EUR -11.2±0.8%, -21.4±0.7% and -28.8±0.8% for RCP2.6, RCP4.5 and RCP8.5 respectively) and an increase in organic aerosol concentration (for EUR +15.1±1.2%, +22.7±1.3% and +34.9±1.3% for RCP2.6, RCP4.5 and RCP8.5 respectively). A slight increase in sea salt particles (for EUR +0.8±0.05%, +1.4±0.06% and +0.2±0.06% for RCP2.6, RCP4.5 and RCP8.5 respectively) and in sulfates (for EUR +4.50±0.62%, +3.0±0.6% and +1.6±0.6% for RCP2.6, RCP4.5

and RCP8.5 respectively) is found as well. An interesting fact about sulfates is that its concentration shows an increase in all scenarios compared with historical simulations, but this happens in a reverse order compared to temperature increase (i.e. RCP2.6>RCP4.5>RCP8.5). This phenomenon will be discussed later in this section. While a decrease in dust particles is observed for RCP4.5 and RCP8.5 (for EUR -5.6±2.4% and -5.3±2.2% respectively), these particles stay nearly constant in RCP2.6. A decrease is seen for the ammonium particles in all scenarios (for EUR -2.1±0.2%, -6.4±0.2% and -10.6±0.2% for

RCP2.6, RCP4.5 and RCP8.5 respectively). The main driving force for the decrease in the concentration of total $PM_{10}$ (seen in figure 4) for RCP4.5 and RCP8.5 scenarios is the decrease of nitrates. The increase in the concentration of other species, especially BSOA, is compensated by the decrease in nitrate concentrations in the case of these two scenarios with a seasonal dependence. As for RCP2.6, in general, since the decrease in nitrates is the lowest among all future scenarios, the slight increase in BSOA, sulfates, dust and salt particles drives the increase in $PM_{10}$ concentrations (figure 7).

In the Mediterranean area, however, the predictions appears to be quite different: in general, the aerosol burden over the Mediterranean area (MED) is higher than that of the European area (average of $9.8 \mu g.m^{-3}$, $9.5\ \mu g.m^{-3}$ and $9.4\ \mu g.m^{-3}$ for EUR compared to $17.4\ \mu g.m^{-3}$, $16.7\ \mu g.m^{-3}$ and $16.6\ \mu g.m^{-3}$ for MED for RCP2.6, RCP4.5 and RCP8.5 respectively, figure 7). This is mainly due to sea salt (approximately $9\ \mu g.m^{-3}$ in all scenarios) and dust (nearly $4\ \mu g.m^{-3}$ in all scenarios). Note that the MED region by definition only contains grid cells over sea, which explains the large sea salt contribution. On the other

hand, for this area, the concentrations of aerosols that depend on continental emissions are considerably lower (-25%, -33% and -32% for black carbon, POAs and ammoniums in MED compared to EUR for RCP4.5 in $PM_{10}$). Nitrates also show a significantly lower concentration here (from $1.03\ \mu g.m^{-3}$ for EUR to $0.21\ \mu g.m^{-3}$, $0.39\ \mu g.m^{-3}$ and $0.25\ \mu g.m^{-3}$ for RCP4.5 in $PM_{10}$ for MED, MEDW and MEDE respectively). Sulfur emissions from maritime shipping lead to sulfate concentrations over the Mediterranean area and especially over its eastern part ($2.5\ \mu g.m^{-3}$ in MEDE compared to $1.99\ \mu g.m^{-3}$ in EUR for

RCP4.5 in $PM_{10}$). Finally, it is worth noting that the BSOA fraction is lower over the Mediterranean (-36%, -31% and -23% of BSOA compared to EUR in MED, MEDW and MEDE respectively).





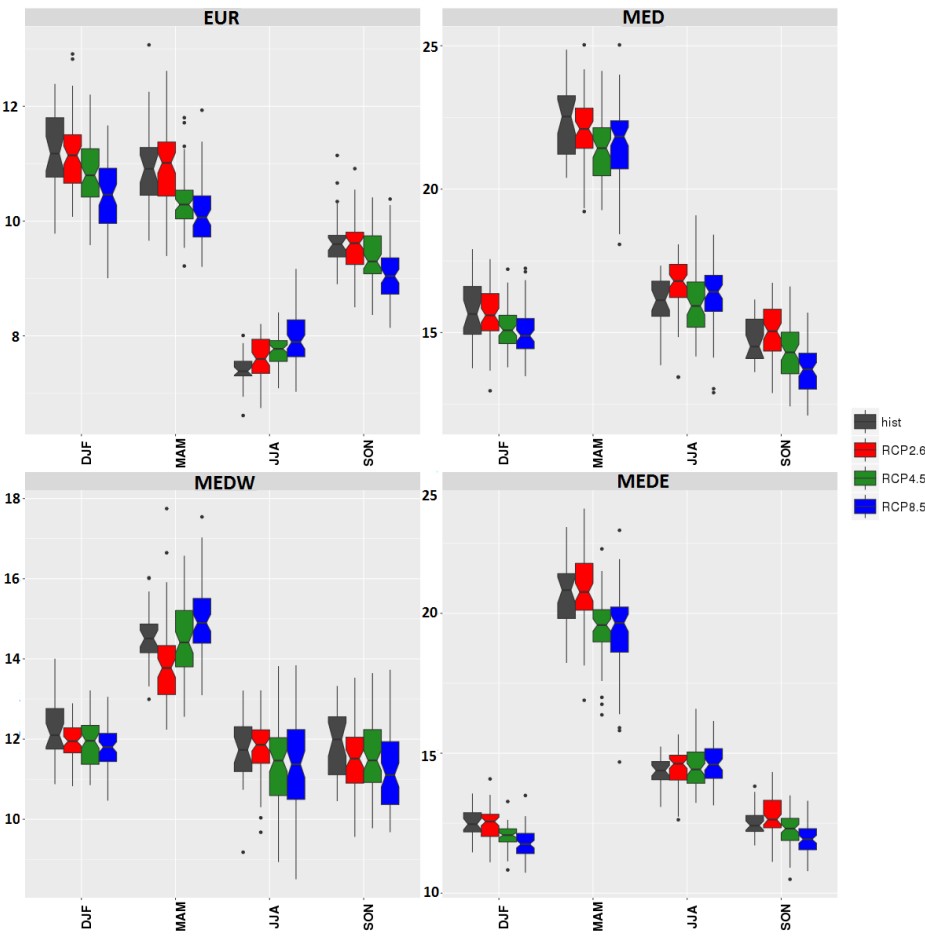

**Figure 6.** Seasonal boxplots for all scenarios for all seasons for $PM_{10}$ (same color code as figure 5) for different sub-domains. Scales are different for different sub-domains.

While figure 8 shows that nitrates, sulfates and dusts show a high seasonal variability, sulfates are quite stable along the year. 2D concentration fields for nitrates, BSOA, sulfates and dust particles are shown in figure 9, for the season when each component shows its highest concentration. This corresponds to the summer period for sulfates and BSOA, and winter and spring for nitrates and dust.

### 3.4 Dependence of $PM_{10}$ components to meteorological parameters

In order to explain the evolution of PM components under future climate scenarios, they are correlated here to different meteorological parameters. The ones tested here are temperature, wind speed, precipitation, relative humidity, planetary boundary layer height and shortwave radiation. Because of the variations in the seasonality of the different PM components, the analysis of their dependencies upon meteorological parameters must be conducted for each season separately. The correlations for all



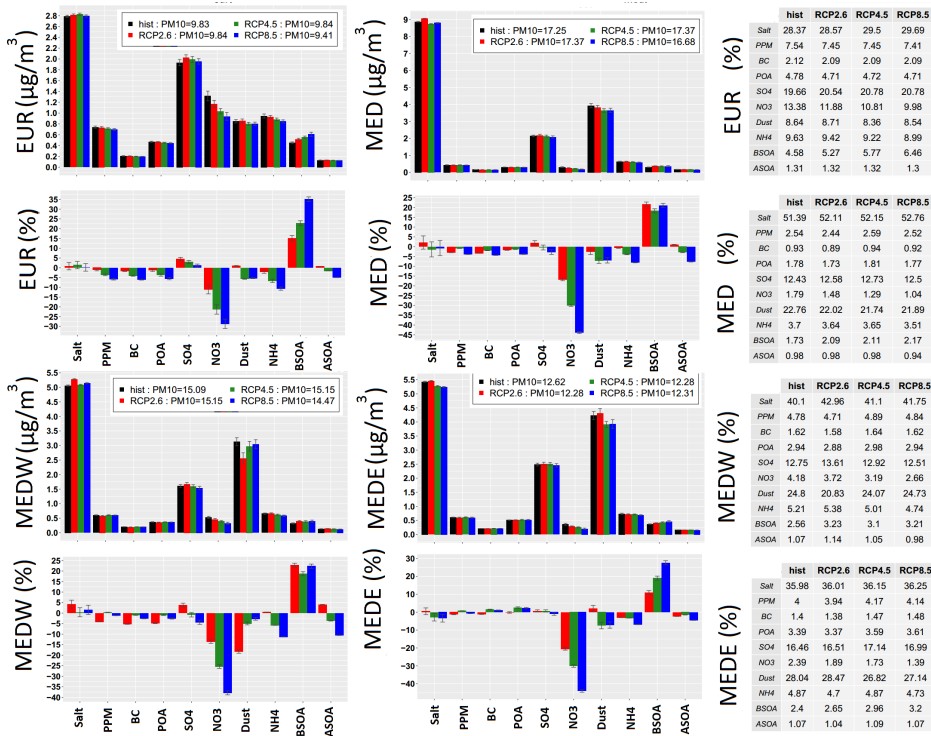

**Figure 7.** Concentrations and relative changes of $PM_{10}$ components (same color code as in figure 5) for all sub-domains averaged over the whole period of simulations in climate change scenarios and for the different sub-domains. Error bars show the confidence interval calculated for annual averages for each sub-domain. The changes show (future-historic)/historic*100. Tables report the percentage of each component in each scenario. The concentration of $PM_{10}$ in total is written in the legend above each figure.

seasons and for the five selected meteorological parameters with $PM_{10}$ components are shown in figure 10 and figure 11 for EUR and MED sub-domains for different years. This makes up to 30 pairs of values for the historic period and 70 for the future period. Here, a Pearson correlation coefficient of greater than 0.6 is considered to account for a significant relationship. Correlations between the different meteorological parameters themselves are shown in SI6 fig.1 and SI6 fig.2. SI6 fig.3 shows the
3-dimensional correlations for the same species as figure 8, with the parameter that correlates best with them for selected seasons when their concentration is highest (i.e. nitrates, BSOA, sulfates and dust with temperature, temperature, relative humidity and PBL height for winter, summer, summer and spring respectively).

### 3.4.1   Inorganic PM components

Particle nitrate concentrations appear to be strongly anti-correlated with temperature (figure 10). That is why, in most regions, they show a decrease in the future scenarios, mainly because of the higher temperature predicted in those scenarios, which might
lead to a shift in the nitrate gas-aerosol partitioning towards the gaseous phase, and more volatilization of already formed nitrate



aerosols. Especially during the winter season, anti-correlations are seen with wind speed, precipitation, boundary layer height, and a correlation with surface radiation. This fits into the picture of a switch from anti-cyclonic conditions - characterized by cold continental weather, clear skies and high solar radiation, large vertical stability and low boundary layer height - to marine conditions. Continental conditions during this season indeed favor enhanced nitrate concentrations while marine conditions are

related to lower concentrations.

The correlation coefficient between nitrates and temperature is the lowest in summer. If we stay again in our synoptic scale framework, hot summer days favor pollution build-up and accumulation, but decrease the partitioning in favor of the particle phase, so these effects compensate each other. In spring, relative humidity shows high correlation with nitrate along-side temperature because higher relative humidity favors nitrate partitioning into the particulate phase, in particular if it exceeds

the deliquescence point of ammonium nitrate (RH > 50%). These hypotheses are supported by the correlations presented in SI4, showing the correlations between different meteorological parameters. For the MED region, a strong anti-correlation is still observed especially for winter and spring. However, the (anti)correlations with other meteorological parameters are less pronounced and not necessarily in the same direction as for EUR, because the distinction between continental and maritime conditions is not valid for this region. Overall, the analysis suggests that the major point to be taken into account for particle

nitrates is their high anti-correlation with temperature (seen in figure 10). This confirms the results seen in Dawson et al. (2010), Jiménez-Guerrero et al. (2012) and Megaritis et al. (2014), who conducted sensitivity studies to individual meteorological parameters.

Sulfates are the second most abundant species in Europe after sea-salt, and the third most important species after sea-salt and dust over Mediterranean. They show an increase in all the future scenarios compared with historical simulations, but in

the inverse order of the degree of severity projected temperature increase (i.e. the increase is strongest for RCP2.6 (5%) and less pronounced for RCP8.5 (1.2%) as seen in figure 7). The spatial distribution of this species varies quite strongly between RCP2.6 scenario and RCP4.5 and RCP8.5 scenarios (figure 9). For instance, in RCP2.6, sulfate increases with respect to historic in the SW part of the domain, while it decreases for the same area in RCP4.5 and RCP8.5. 2D correlation maps of sulfates with relative humidity show a strong correlation between these two parameters, especially for the Mediterranean and

for the Atlantic, but also for the EUR sub-domain at winter and spring periods (SI6 fig.3). The positive relationship between sulfates and RH could be related to two different processes. First, during the winter half year, the major pathway of $SO_4^{2-}$ formation from SO2 proceeds via aqueous chemistry, and large scale RH values are a tracer of sub grid scale cloud formation (Seinfeld and Pandis, 2016). On the contrary, during summer, over MED, gas phase $SO_4^{2-}$ formation via SO2 oxidation by OH is dominant, and increased future RH levels, along with increasing temperatures, may lead to increased OH levels (Hedegaard

et al., 2008). However for summer and fall, the PBL height shows more correlation with this pollutant, which is shown in figure 10. Another parameter that shows a strong anti-correlation with sulfate concentrations is the wind speed in spring and winter periods (figure 10).

Ammonium concentrations show a steady decrease in all future scenarios and in all sub-domains. Correlations of $NH_4^+$ with meteorological variables appear to be a combination of those simulated for $SO_4^{2-}$ and $NO_3^-$ with whom $NH_4^+$ forms inorganic





aerosol. When looking at correlations, a relationship is seen for ammonium with relative humidity and wind speed, as well as a strong correlation with PBL height (figure 11).

### 3.4.2  Biogenic SOA

BSOA concentrations show a steady increase in future scenarios for the European sub-domain (figure 8 and 9). While the increase is seen in all sub-domains, their intensity and the scenario dependence are not the same. The formation of the biogenic

organic aerosol fraction greatly depends on its precursors (isoprene and mono-terpenes), which emissions globally increase with temperature. In EUR, isoprene emissions increase by 20.3%, 31.1% and 52.5%, and mono-terpenes emissions by 15.7%, 24.0% and 38.1% respectively for RCP2.6, RCP4.5 and RCP8.5. There are many studies that investigated the changes of BVOC emissions in the future. However, not many of them took into account the climate effects only. For example, Lathière et al., (2005) used the full version of MEGAN (which includes $CO_2$ inhibition and the dependence of isoprene emissions upon ozone

concentrations) to calculate a 27% and 51% increase in isoprene and mono-terpenes respectively in 2100 compared to 1990s with a scenario that is close to RCP4.5. This is rather similar for isoprene, but more than the double for mono-terpenes, as compared to our study. Pacifico et al. (2012) calculated a 69% increase in isoprene with a RCP8.5 scenario in 2100 compared to 2000s, Hantson et al. (2017) found 41%/25% ratio in 2100 compared to 2000 for isoprene/mono-terpenes respectively with RCP4.5. Langner et al. (2012) explored four different models for the European region (DEHM, EMEP, SILAM and MATCH)

finding an isoprene increase in the range of 21%-26%, the increase in isoprene emissions in our simulations for the same period amounts to 21%. These results are consistent with our simulations. In general, isoprene and mono-terpenes emissions show high sensitivity to temperature, $CO_2$ inhibition (Arneth et al., 2007 ; Young et al., 2009 ; Tai et al., 2013) and land-use changes, which makes their estimation for future scenarios highly uncertain.  This increase in precursors results in an increase in production of BSOAs (for EUR 15.1%, 22.7% and 34.9% for RCP2.6, RCP4.5 and RCP8.5 respectively), with

similar effect predicted in other studies (Heald et al., 2008 ; Megaritis et al. 2014). However, higher temperatures induce higher evaporation for semi-volatile organic compounds, therefore, lower formation of organic aerosols is to be expected. This fact has been shown in Dawson et al. (2007), where without changing biogenic emissions, the temperature was increased by 5°C and a decrease of almost 20% was seen for SOA production over the eastern US. In the case of our simulations, the increase in biogenic precursors resulting to formation of biogenic SOA trumps the evaporation effect due to higher temperature, therefore

an increase in biogenic SOA is seen. Figure 10 shows the correlation coefficient of BSOA with temperature for the EUR and MED sub-domains (SI6 fig.3 for spatial correlations). For the EUR sub-domain, statistically significant regressions are seen for all scenarios during summer for temperature and also for shortwave radiation. For the Mediterranean sub-domains, no mono-terpenes or isoprene is emitted from the sea surface, therefore the concentration of BSOA in the Mediterranean area is the result of transport from the continental area, which results in low correlations between temperature and BSOA for the

Mediterranean sub-domains, while a high correlation is found between wind speed, relative humidity and PBL height. Another fact that seems to be interesting to mention is the percentage of the production of BSOA from isoprene and mono-terpenes; year-long sensitivity tests (reduction of terpene and isoprene emissions each by 10% in two separate one year long simulations) for the year 1998 with historic climate show a distribution of 21% and 79% annually and 39% and 61% for the summer period




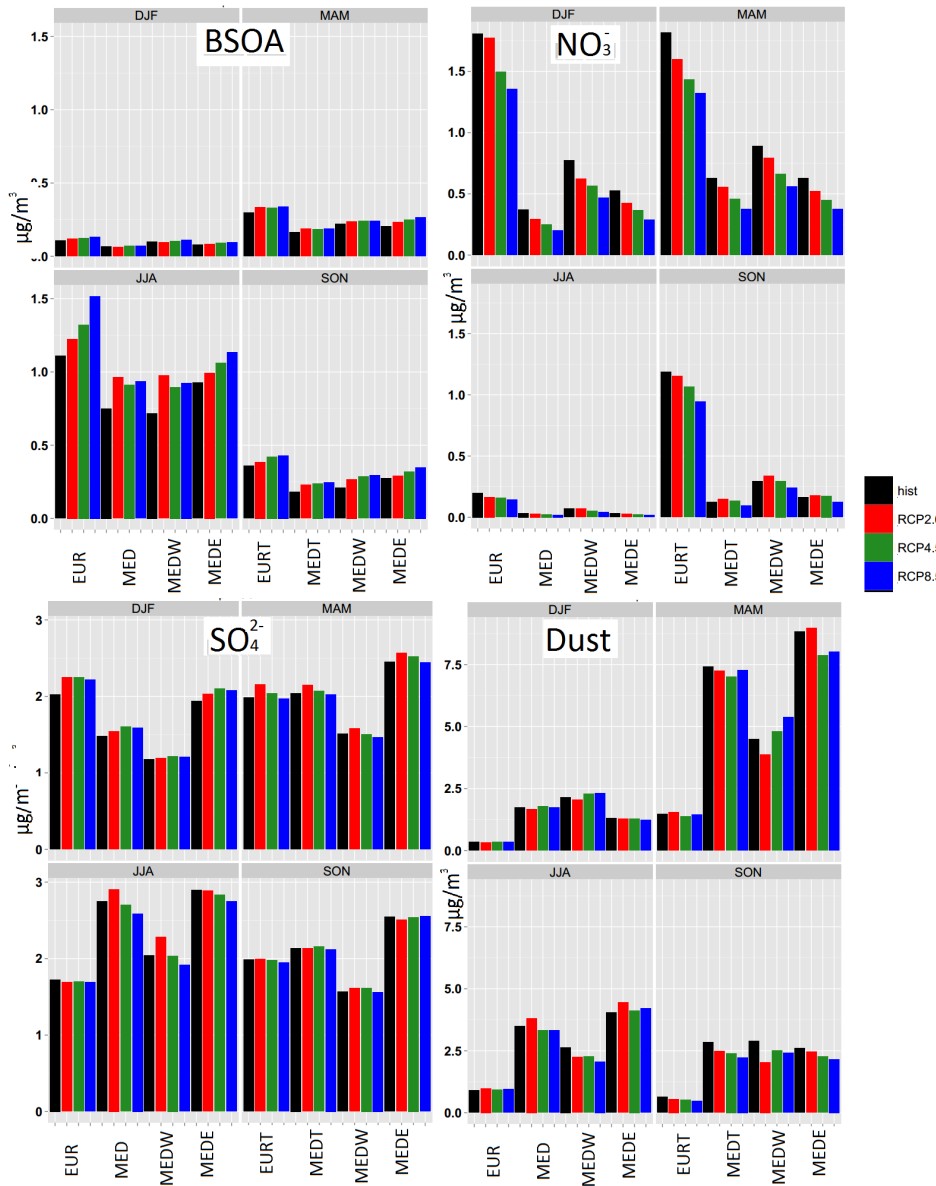

**Figure 8.** Seasonal absolute concentrations of BSOA, nitrate ($NO_3$), sulfate ($SO_4^{-2}$) and dust particles (same color code as in figure 5) for different sub-domains. Each panel shows one of the species mentioned above for four scenarios and each sub-panel shows one season.

for the production of BSOA from isoprene and mono-terpenes respectively. The isoprene to mono-terpene emissions ratio is 2.9 annually and 6.1 for the summer. Similar results were seen in Aksoyoglu et al., (2017).



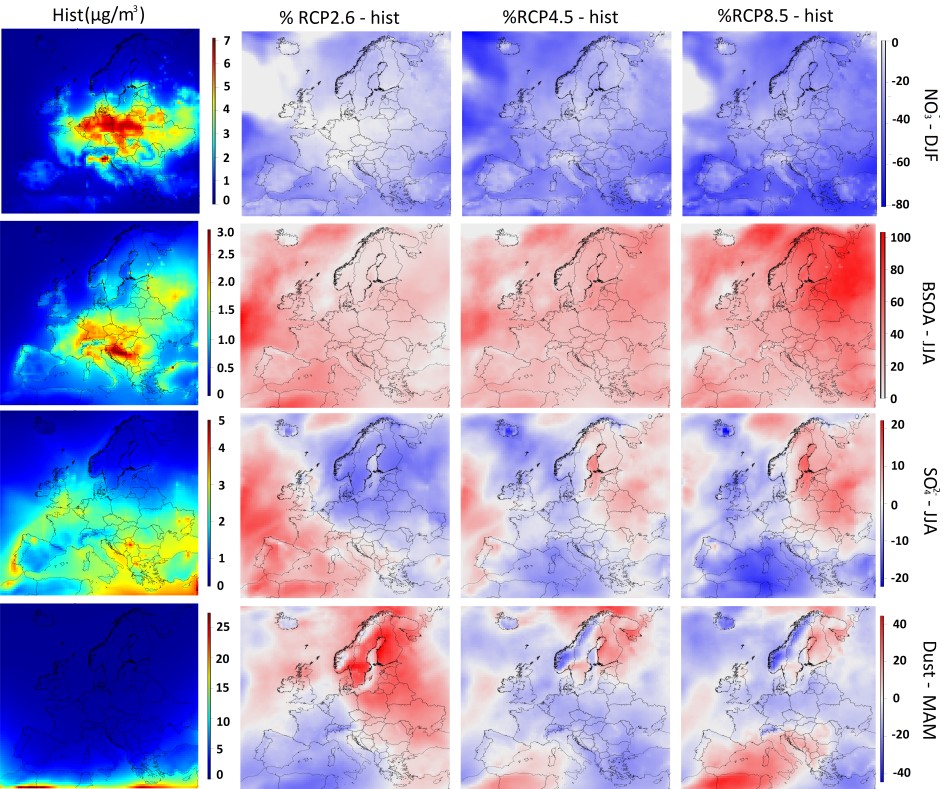

**Figure 9.** Historic concentrations and relative changes in future scenarios for biogenic SOA (BSOA), nitrates ($NO_3^-$), sulfate ($SO_4^{2-}$) and dust particles for selected season in which the concentrations are the highest. Figures on the left column show the average concentration of historic simulations, the three columns on the right show the relative difference of RCP2.6, RCP4.5 and RCP8.5 future scenarios respectively to historic simulations. Each row shows one season, scales are different for each row of simulations.

### 3.4.3 Dust and salt particles

Dust concentrations are predicted to be at their maximum during spring (figure 9). This phenomenon is also described in the literature for different regions (Werner et al., 2002 and Ginoux et al., 2004 exploring global simulations, Laurent et al., 2005 for China and Mongolia as well as Vincent et al., 2016 for the western Mediterranean). In our simulations, spatial maximum in

dust particle concentrations normally occurs in the eastern Mediterranean (figure 8, 9 and 10). While a decrease is observed for RCP4.5 and RCP8.5 (figure 7, for EUR -5.6% and -5.3% respectively), these particles do not exhibit any sensible variations in RCP2.6. Since dust emissions are not taken into account within the simulated domain in our simulations and boundary conditions are the same in these scenarios, changes in advection of dust aerosols is responsible for these variations. Therefore only advection plays a role, and cannot be captured by a local correlation analysis by definition.

The reason for the increase of dust particles in RCP2.6 scenarios for MEDE can be the different behavior that is seen in meteorological parameters for this sub-domain compared to the others. Its temperature decreases after 2040s in contrast to the



other two scenarios, and therefore induces changes in relative humidity that are different from the other scenarios. The average of RH in RCP2.6 stays higher than the other scenarios, while it is lower than historic simulations for MEDE (SI3 fig.2). Up to now, we mostly have discussed climate change related modifications in PM sources, some sinks and transport. We also should

take into account the effect of climate related changes effecting wet deposition. For RCP4.5 and RCP8.5, precipitation amount stay rather constant over the MEDW, while it increases in RCP2.6 (SI3 fig1). Precipitation duration decreases for RCP4.5 and 8.5 over this sub-domain, while it stays quite constant for RCP2.6 for MEDW. Thus both precipitation amount and duration is stronger in RCP2.6 than in RCP4.5 or RCP8.5. This could, in addition to different advection patterns, explain the lower MEDW dust concentrations in RCP2.6. However, for MEDE, precipitation amount increases as the number of rainy hours

decreases, which shows strong, but non-frequent rain episodes, which would explain the increase in the dust concentrations in this sub-domain.

Salt particles show a high concentration in the Mediterranean area and also in the Atlantic Ocean, they are the most important PM species for the EUR and MED sub-domains. Sea salt emissions are very sensitive to wind speed, leading to a correlation between salt concentrations and wind speed (figure 11). Therefore, the small changes of salt concentration in future scenarios

compared to historic simulations are mostly because of small wind speed changes in future climate.

### 3.4.4  Total $PM_{10}$ and $PM_{2.5}$ dependencies to meteorological components

Investigating the correlation between $PM_{10}$ and $PM_{2.5}$ and different meteorological parameters, reveals high spatial and temporal variability. For the EUR sub-domain, for both $PM_{10}$ and $PM_{2.5}$ the PBL height parameter shows the highest correlation (anti-correlation). For the Mediterranean region, among all the investigated meteorological parameters, $PM_{10}$ seems to be more affected by wind speed and $PM_{2.5}$ more by relative humidity. The analysis of the link between total $PM_{10}$ and total $PM_{2.5}$ with meteorological parameters (figure 11) is far less conclusive compared to individual component analysis. As a generalized conclusion, $PM_{10}$ and $PM_{2.5}$ tend to follow the correlations of their largest contributor.

Finally, precipitation has been pointed out as a crucial, but difficult to predict parameter in future climate scenarios (Dale et al., 2001). In our study, the correlation of PM or PM components with precipitation amount are generally weak (and sometimes even correlations are seen instead of expected anti-correlations). It has also been discussed, that precipitation duration could be more impactful on the PM than the amount (Dale et al., 2001). No correlation study has been performed with this parameter in this work, but the decreasing precipitation duration in all scenarios could induce some increase in PM and its components.

## 4  Impacts of boundary condition and anthropogenic emissions

While climate on its own can have important impacts on the future concentrations of different species, other drivers might have their specific effects on PM concentrations, which can either amplify or compensate the climate-related effects. This section explores the impacts of two other drivers: boundary conditions and anthropogenic emissions. Five sets of simulations have been used to achieve this goal, one where boundary conditions were changed together with climate inputs, and two

where anthropogenic emissions were changed alongside climate inputs; these simulations have been compared with historic




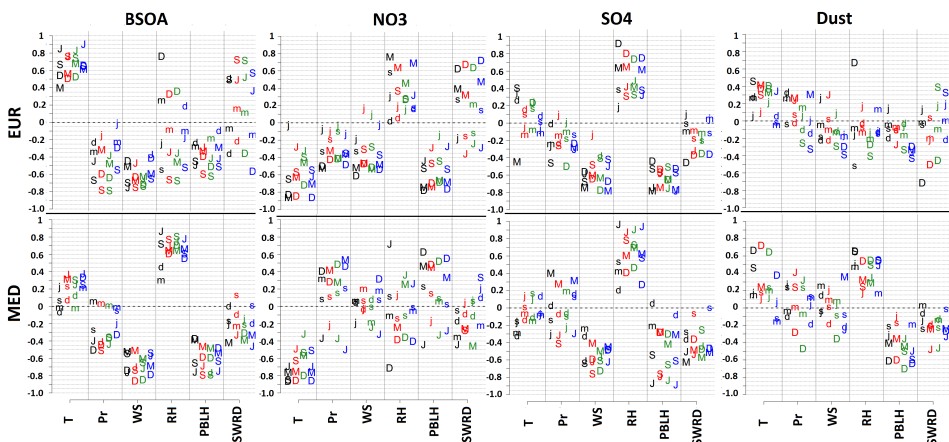

**Figure 10.** Correlation coefficient between all tested meteorological parameters to BSOA, (NO$_3$), sulfate (SO$_4^{-2}$) and dust particles for all seasons and EUR and MED sub-domains. D, M, J and S represent winter, spring, summer and fall (first letter of the first month of each season). Up%ase letters mean that the correlation between the two parameters is statistically significant, lowercase letters show the contrary. Color coding for different scenarios is the same as previous figures.

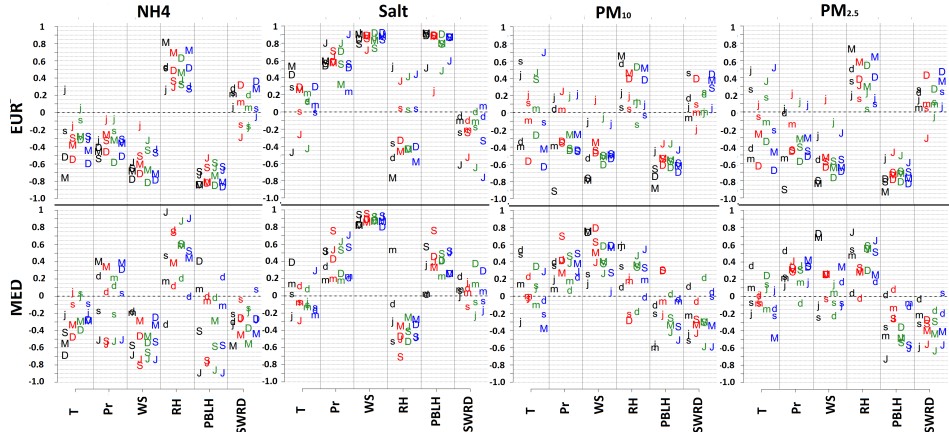

**Figure 11.** Correlation coefficient between all tested meteorological parameters to Ammonium (NH$_4^+$), salt, PM$_{10}$ and PM$_{2.5}$ particles for all seasons and EUR and MED sub-domains. D, M, J and S represent winter, spring, summer and fall (first letter of the first month of each season). Up%ase letters mean that the correlation between the two parameters is statistically significant, lowercase letters show the contrary. Color coding for different scenarios is the same as previous figures.

simulations as well as climate simulations with constant boundary condition and anthropogenic emissions. At last, in an effort to give a more comprehensive view of what the accumulative effects of all drivers would be, the aforementioned simulations will be compared to a series of simulations where all drivers change at the same time. In these simulations, RCP4.5 related climate and boundary conditions are compared to the historic ones, 2050 regional current legislations emissions (CLE) and



maximum feasible reductions (MRF) are compared against 2010 emissions. In total, in this section, six series of simulations are presented (the numbers inside parenthesis refer to table 1): historic simulation (simulation #1), climate impact simulation (simulation #3), boundary condition impact (simulation #5), emission impact (simulations #6-7) and accumulative impacts (simulation #8)). For each series, 10-year long simulations are used, between 2046-2055 for future scenarios and 1996-2005 for historic simulations. In this way, the effect of boundary conditions, emissions and climate were calculated separately and

compared with the overall changes of simulation #8. In figures 12 the impact of each driver is shown separately, in terms of relative change (SI8 for seasonal changes).

## 4.1 Boundary conditions

Among $PM_{10}$ components, dust, nitrate, BSOA, primary organic aerosols (POA) and sulfates show the highest impact on future PM concentrations when boundary conditions change, the other species showing only minor changes or no change at all (figure

12). Among these species, dust particles show the strongest dependence on boundary conditions, that is an increase of +77±2%, 30±10.7%, 9±1.9% and 51±15.2% for EUR, MED, MEDW and MEDE for future RCP4.5 with respect to historic boundary conditions. Their simulated dependence on climate was indeed much smaller (-9±0.3%, -4±1.4%, +3±0.6% and -4±1.8% for EUR, MED, MEDW and MEDE for the same period). It is important to bear in mind that the concentration of dusts in the European sub-domain in our simulations is on average $0.8 \mu g.m^{-3}$ with an important spatial variability, concentrations dropping

significantly as we go further north (figure 9). Therefore, the low relative changes simulated for the Mediterranean sub-domains have to be considered as the sign of a high absolute sensitivity to the scenario. The reason for the important changes in these species can be due to wind intensity and humidity in source regions and on the other hand land-use changes caused by climate change outside our domain. There are many uncertainties regarding the future changes of the dust concentrations (Tegen et al., 2004; Woodward et al., 2005). Changes in climate drivers such as precipitation, wind speed, regional moisture balance in

source areas, and land use changes, either resulting from anthropogenic changes or climatic reasons can have important effects on dust emissions (Harrison et al., 2001). Projection of changes in land use resulting from both sources are highly uncertain, which results in strong uncertainties in projections of dust concentrations changes in future scenarios (Tegen et al., 2004; Evan et al., 2014).

Nitrates and BSOA are more affected by climate change impact than by boundary condition input changes both for all

sub-domains (-13±2.6%, +14±0.5%, -2±0.8% and +12±0.7% for RCP4.5 compared to -6±2.5%, -26±0.5%, -23±0.9% and –25±0.7% for boundary condition changes for nitrates and +12±1.3%, +23±1%, +24±1% and +12±1.1% for RCP4.5 compared to +6±1.3%, +14±1.3%, +7±1% and +14±1.4% for BSOA for EUR ,MED, MEDW and MEDE respectively).

Contrary to the species discussed above, sulfates and POA are more sensitive to boundary conditions than to climate effects. Sulfates show a -2±1.2%, (-7±1.6%, -6±0.9% and -7±1% decrease. POA shows -9±0.7%, -26±0.1%, -11±0.1% and -19±0.1%

of decrease related to boundary conditions for EUR, MED, MEDW and MEDE respectively, while climate effects on these particles were negligible.





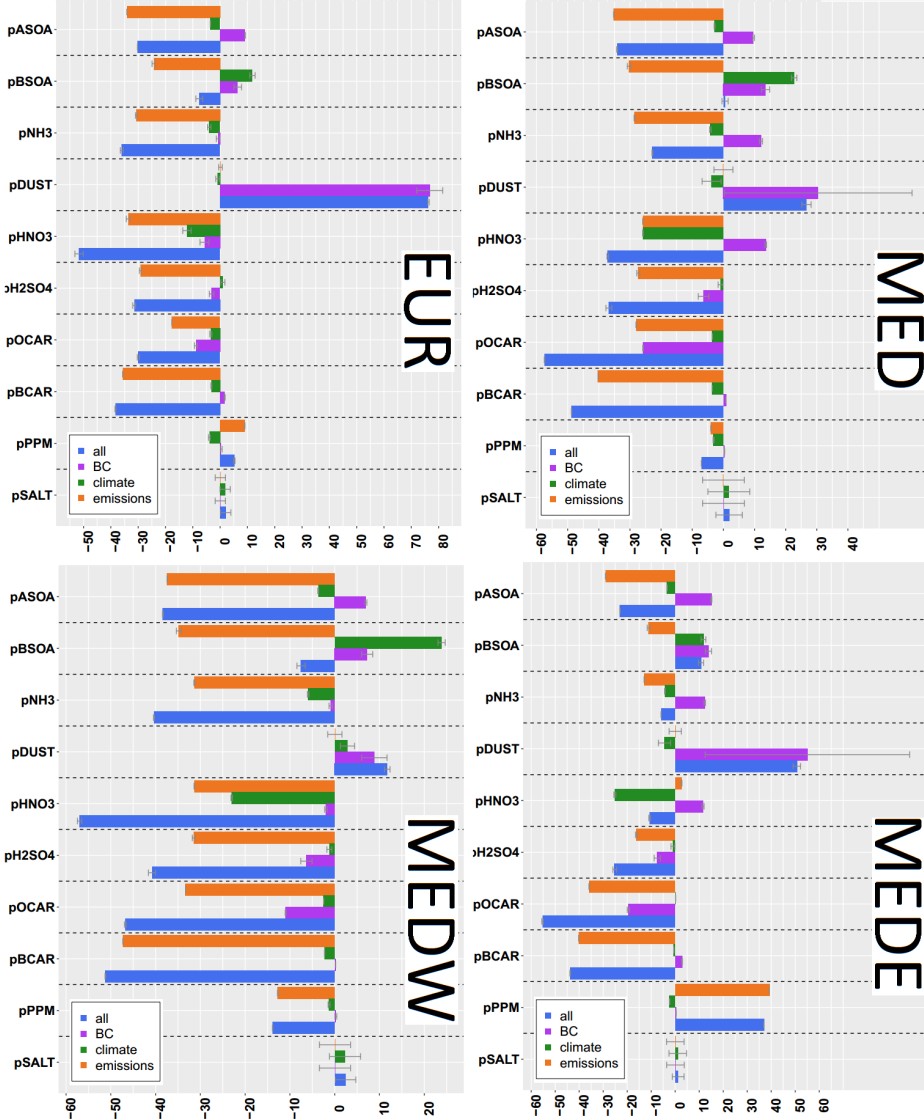

**Figure 12.** Relative impact of climate, boundary conditions and emission drivers on $PM_{10}$ components for different sub-domains. Error bars show the confidence interval calculated on annual averages.

Ammonium shows a negligible change with regards to boundary conditions in EUR, and an increase in most Mediterranean sub-domain (+12±0.3%, -1±0.1% and +12±0.2% for MED, MEDW and MEDE), while their change was mostly negative for climate impact results.



## 4.2 Anthropogenic emissions

Changes in anthropogenic emissions in CLE and MFR 2050 inputs are compared to CLE 2010 emissions for different species in SI (SI7 fig.1). As expected, a decrease is seen for most species in CLE 2050 emissions, but to a higher extent in the MFR 2050 ones. A simple comparison between CLE and MFR 2050 emission scenarios is shown in figure 13. Bear in mind that this figure shows the effects of climate change (RCP4.5) and emission change at the same time, so does every value presented in this paragraph. For CLE simulations, sulfates show a decrease of -28±1.2%, -29±1.3%, -34±0.9% and -18±0.9% for EUR, MED, MEDW and MEDE respectively with respect to historic simulations, while for MFR scenarios, they show a -60±1%, -51±1.1%, -55±0.8% and -52±0.7% decrease for the same order of sub-domains. The reason for this decrease is the decrease in the emissions of SO2 (SO2 emissions reduction of -30%, -53%, -52% and -42% for CLE and -60%, -53%, -57% and -68% for MFR for the same order of sub-domains). Particle nitrates show a strong decrease with decrease of precursor emissions as well, presenting -48±2.5%, -32±0.4%, -36±0.8% and -28±0.6% for CLE and -79±2.2%, -61±0.4%, -74±0.8% and -77±0.5% for MFR for EUR, MED, MEDW and MEDE respectively (NOx emissions reduction of -60%, -38%, -48% and -30% for CLE and -84%, -38%, -76% and -76% for MFR for the same order of sub-domains). Ammonium shows the same behavior, showing -36±1%, -55±0.2%, -56±0.1% and -18±0.1% for CLE and -68±0.9%, -87±0.2%, -87±0.1% and -64±0.1% for MFR for EUR, MED, MEDW and MEDE respectively. Other components such as BC and POA show strong decrease since their concentrations depend directly on the amount of anthropogenic emissions. Interestingly, BSOA concentrations show a strong decrease related to changes in anthropogenic emissions (contrary to the increase for climate change alone). A -7±1.3%, -34±1.1%, -39±1.1% and +4±1.3% for CLE and -36±1.1%, -64±1%, -65±0.9% and -35±1.1% for MFR for EUR, MED, MEDW and MEDE is seen respectively for this species. The fact that the decrease in anthropogenic emissions overshadows the increase in BSOA when looking at climate and boundary condition effects is an important message to take away from these simulations. The exact mechanism of this effect is not clear, it could be due to the general decrease of seed aerosol in these scenarios modifying the gas/particle equilibrium for SVOC (semi volatile organic compounds) formed from mono-terpenes and isoprene oxidation. Changes in oxidant levels because of decrease in anthropogenic emissions and also a direct decrease in anthropogenic COVs are other reasons for this change (i.e. less OA mass available for oxidation products to condense on).

For the comparison of driver impacts (figure 12), only CLE 2050 simulations for emission impact scenarios are used since CLE 2050 emissions are used in the simulation where all drivers change (to be presented later). Almost all species show strong dependence to emission changes, except dust and salt particles. Quantitative effects vary for the Mediterranean sub-domains: the effect of emission changes becomes less pronounced for species without maritime emission sources (such as $NH_4^+$), while they stay high for species like POA, BC and $SO_4^{2-}$ which can be emitted by shipping lines.

## 4.3 Cumulative impacts

To provide a more complete view of the 2050's probable atmospheric composition (under the hypotheses of the scenario), the "all" scenario (Figures 12) shows the combined effects of all drivers changing at the same time. There are many uncertainties affecting future scenarios as it can be surmised; but with regard to drivers that are being explored here, this scenario shows



what a more realistic future air composition might look like. As seen in figure 12, and with regard to total $PM_{10}$ and $PM_{10}$ components, the changes in emissions set the tone for the future, meaning that reduction in anthropogenic emissions, overshadows the climate and the boundary condition drivers for most of species, pointing out that mitigating air pollution with respect to air quality in the future depends greatly on the reduction of anthropogenic emissions.

For $PM_{10}$ and for the period of 2046-2055 with RCP4.5, the different drivers indicate a decrease of -15.6%, -6.7%, -10.5% and +4% for EUR, MED, MEDW and MEDE respectively mainly driven by anthropogenic emissions. Because of the boundary condition changes, PM increases of +5.3%, +6.8%, +1.2% and +15.1% for EUR, MED, MEDW and MEDE respectively are observed, mainly because of dust concentration increase. The climate impacts on $PM_{10}$ concentrations for the same period are -2.9%, -0.5%, +0.6% and -1.6% for EUR, MED, MEDW and MEDE respectively. The total change for the period of 2046-2055 and RCP4.5 (meaning in simulations where all drivers change at the same time) a decrease is seen for all sub-domains for $PM_{10}$ (-11.8%, -1% and -9.2% for EUR, MED MEDW, and MEDE respectively) except for MEDE where an increase of 9.1% is seen. Thus, for most of the domains, the effect of emission reductions on $PM_{10}$ concentrations around 2050 is to a certain extent reduced by modifications of boundary condition and regional climate.

## 5 Conclusion and discussion

We investigated the effect of different drivers on total $PM_{10}$ and $PM_{10}$ components in future scenarios for different sub-domains. For this purpose, an exhaustive number of scenarios plus historic simulations were performed. The drivers that are taken into account include climate change, anthropogenic emissions and boundary condition changes. For each driver, simulations are compared against historic simulations and then the effect of a specific factor is calculated separately and compared to a scenario where all drivers change at the same time. This approach is taken since in the existing climate change literature, the effects of different drivers are taken into account either all at once (Lacressonnière et al., 2017), separately but for a single driver and for a short period of time (Heald et al., 2008), separately but using sensitivity tests (Dawson et al., 2007; Megaritis et al., 2014), or in the best of cases, separately and for an acceptable period of time but for only one driver (Lemaire et al., 2016). The goal of this work was to explore multiple drivers separately for $PM_{10}$ and its components for a coherent and comparable set of future scenarios, therefore making climate change analysis more comprehensible and finding the drivers with the most impact on the $PM_{10}$ future concentration changes easier. This work focuses on the Mediterranean area, since there have not been many studies focusing on the climate change drivers in the Mediterranean area, although this region might be highly sensitive to climate change.

Future scenarios that we performed show that in the 2050s, in the case of an RCP4.5 scenario and CLE 2050 emissions, a general increase in temperature and a decrease of $PM_{10}$ total average concentration is seen both in the European sub-domain (-12%) and the Mediterranean one (-1%). The diminution of $PM_{10}$ has also been reported in the literature in other studies for the European sub-domain (e.g. Markakis et al., 2014; Lacressonnière et al., 2014; Lacressonnière et al., 2017), the intensity of this decrease changes with the period that was taken into account and also the inputs used. The PM changes are far from uniform for different seasons: a maximum change of -24% for spring and +9% for summer for the European area. For the



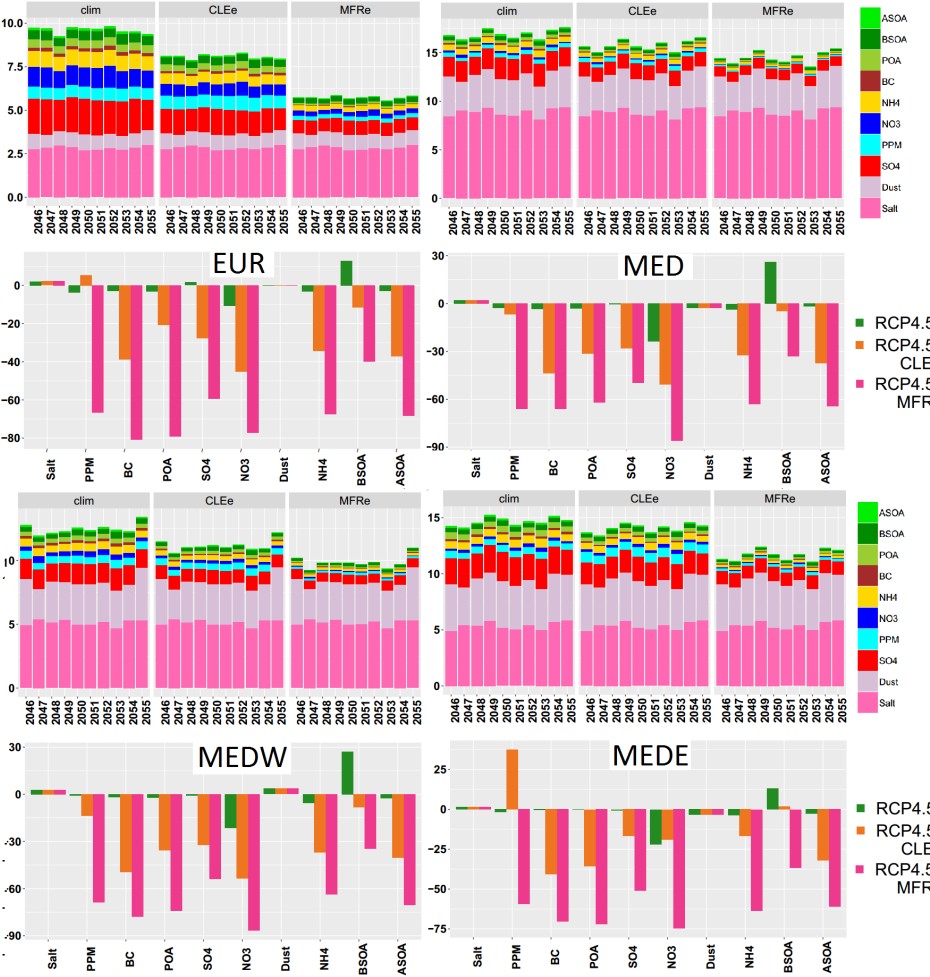

**Figure 13.** Emission scenario compariosons. Each panel shows one sub-domain, the upper sub-panels show the $PM_{10}$ components for each scenario, the lower sub-panel shows the percentage of difference for each scenario ((future – historic)/historic*100).

Mediterranean area, a maximum change of +25% for winter and -19% for spring are observed. These values seem to indicate different behaviors observed in the Mediterranean compared to the European area. The reasons for these changes were explored driver by driver and the effect of each driver was estimated.

Regional climate change alone, results in a decrease of $PM_{10}$ in RCP4.5 and RCP8.5, while RCP2.6 shows an increase for $PM_{10}$. Among the $PM_{10}$ components, BSOA and nitrate particles show the most sensitivity to climate change. It appears that, when exploring the impacts of climate change, nitrate decrease governs the decrease of $PM_{10}$ and $PM_{2.5}$ in RCP4.5 and RCP8.5, however, in RCP2.6, the increase in dust, salt and BSOA particles outweighs the decrease in nitrates. Seeking for reasons for the changes seen for PM components, correlations of meteorological parameters to individual components were investigated. Nitrates show a strong dependence (anti-correlation) to temperature, especially during winter, when a correlation





to shortwave radiation and anti-correlations to wind speed and PBL height are also observed. These relationships seem to suggest a switch to slightly more "marine-type" conditions for a future climate during winter (supported by correlations calculated between different meteorological parameters). BSOA also shows a strong correlation to temperature (and therefore shows a strong increase in future scenarios) in all sub-domains, resulting from an increase in BVOC emissions because of higher temperatures. Sulfate particles are seen to have a correlation with relative humidity and PBL height, the extent of this correlation

changes depending on the sub-domain explored. The relationship of $SO_4^{2-}$ with RH can be related to either production of this aerosol from SO2 or formation of gas phase $SO_4^{2-}$ from oxidation of SO2 by OH, the domination of these processes depend on the sub-domain and the season in which they are studied. Relationship of ammonium aerosols with meteorological conditions is a combination of those of $SO_4^{2-}$ and $NO_3^-$. Salt particles show a clear correlation to wind speed, while dust concentrations present a weak relationship with the tested meteorological parameters since their changes are related to advection and therefore

not captured by local correlation analysis.

Future changes in boundary conditions (depicting long range transport from outside of Europe) greatly affect dust concentrations, especially over the Mediterranean area (see below). On the contrary, they have only limited impact on sulfate, nitrate ammonium and OA concentrations. Emission changes show the largest effect on all $PM_{10}/PM_{2.5}$ components except salt and dust particles. One of the most interesting cases that was encountered in emission change scenarios, was the decrease in BSOA

because of anthropogenic emission changes. This is tentatively attributed to changes in seed aerosol and the changes in oxidant levels because of decrease in anthropogenic emissions and also a direct decrease in anthropogenic COVs. However, the exact mechanism of this relationship still needs further investigated. Compared to the other two drivers, the effects shown by the anthropogenic emission reduction are undeniably more important for most species. This gives us the take-away message that, according to our study, anthropogenic emission reduction policies (or lack thereof) will have a strong impact on the concen-

trations of PM seen in the future and also the fact that its impact will be more significant than that of regional climate and long-range transport.

Another point that has been raised in this article is the differences between the European sub-domain and the Mediterranean Sea on one hand and between western and eastern Mediterranean on the other hand. The behavior of these sub-domains differ when they are exposed to climate change. Meteorological changes in the domain show increasing temperatures, increasing

PBL height and decreasing humidity. Winters and springs seem to become drier and the other two seasons wetter when regarding to precipitation amount, while rain episodes become more intense and shorter in most cases (except RCP2.6). The concentration of $PM_{10}$ in general is higher in the Mediterranean due to higher concentrations of dust and salt particles, while its annually averaged changes in the future stay quite similar to what was seen for the European sub-domain. Seasonally, in the Mediterranean a maximum of $PM_{10}$ concentrations is seen for spring when dust episodes are more common, contrary to the European sub-domain (maximum at winter for EUR). Emission reduction policies will reduce the concentrations of anthropogenic species in the basin by almost the same percentage as the European sub-domain as shown in this work (for example, for sulfates, anthropogenic emissions reduction results in -29% and -30% for EUR and MED respectively for CLE2050 emis-

sions). While this fact shows that emission reduction policies will reduce the $PM_{2.5}$ and lower aerosol fraction pollution, they will not lessen the Mediterranean $PM_{10}$ burden by much, since in this area the $PM_{10}$ concentration revolves around dust and



salt concentrations. For the dust concentrations, our scenarios show an increased concentration in the Mediterranean due to long-range transport, especially in the eastern basin. However, changing land-use in the northern African area will affect the concentration of dust in the Mediterranean, but the extent and even the direction of this change is uncertain, literature suggests

that the dust concentrations due to land-use changes in future scenarios can decrease or increase depending on the scenario that has been taken into account (Tegen et al., 2004; Woodward et al., 2005). The salt particle concentration changes are also uncertain, literature have shown that there seems to be a correlation between rising temperature, rising sea-levels and salinity (Rohling and Bryden, 1992; Tsimplis and Rixen, 2002), but the results of model studies of sea salinity are highly uncertain (Meier et al., 2006).

While exploring the three aforementioned drivers is important to understand the behavior of PM and PM components in the future, there are other aspects that need exploring as well in future studies. Other additions to this study would be for example to explore the effects of land use changes, OA simulation scheme changes and BSOA trend changes related to ASOA changes. Additions to inorganic aerosol formation in future scenarios (such as different salt formation schemes, dimethyl sulfide formation from sea surface...) would be useful additions to the field of climate change study as well. The driver

by driver approach can be taken with each of these parameters in order to explore their effects on future changes of PM concentrations.

*Acknowledgements.* This research has received funding from the French National Research Agency (ANR) projects SAF-MED (grant ANR-15 12-BS06-0013). This work is part of the ChArMEx project supported by ADEME, CEA, CNRS-INSU and Météo-France through the multidisciplinary program MISTRALS (Mediterranean Integrated Studies aT Regional And Local Scales). The work presented here received

support from the French Ministry in charge of ecology. This work was performed using HPC resources from GENCI-CCRT (Grant 2018-A0030107232). R. Vautard is acknowledged for providing the WRF/IPSL-CM5-MR Cordex simulations, and D. Hauglustaine and S. Szopa are acknowledged for providing the INCA simulations. Z. Klimont is acknowledged for providing ECLIPSE-v4 emission projections. The thesis work of Arineh Cholakian is supported by ADEME, INERIS (with the support of the French Ministry in charge of Ecology), and via the ANR SAF-MED project. Giancarlo Ciarelli thanks ADEME and the Swiss National Science Foundation (grant no. P2EZP2_175166).



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
