# Peer review of "Future climatic drivers and their effect on $PM_{10}$ components in Europe and the Mediterranean Sea"

_Atmospheric Chemistry and Physics, 2018_

## Referee Comment (RC1) · Anonymous Referee #1 · 31 Dec 2018

The paper presents a wealth of data on future climate, based on several climate simulations, emission scenarios, and boundary condition scenarios, for a domain covering Europe and the Mediterranean Sea. The data are analyzed in detail, with attention to all contributors to PM10 (PM2.5) and the impact of the different drivers. The analysis is done for the European continent, for the Mediterranean basin and two smaller subdomains covering parts of the Mediterranean area. The problem with the paper is that there is nearly too much information, and that it is difficult to remain focused on the main results. A major point of criticism is that the Mediterranean domain, and more so for the subdomains, are far smaller than the European domain. It is not completely fair to contrast them, and within Europe, different regimes could be found as well (e.g.

[Figure]

EUROCORDEX regions). Also , the Mediterranean are is not even fully covered by the model and significantly affected by the boundary conditions. Nevertheless, the approach as such is not wrong, if one is clear on the limitations, and I would encourage the authors to motivate the choices more clearly. Below, detailed comment is given.

Major comments

Abstract: No discrimination was made in the conclusion for the European area and for the Mediterranean area, whereas a lot of attention was given to the differences in the main text. Add a few sentences on the major differences. Dust is not an anthropogenic emission, so only long-range transport and boundary condition effects can be studied in the present set-up.

P2 38-51: Here the motivation of the choice for Europe and Mediterranean is made, but could be improved. North-eastern Europe and Mediterranean are mentioned as hotspot, but in the next sentence Europe as a whole is mentioned. The relation to Charmex is only mentioned, not made in the paper. Should this come back in the conclusions?

P3 It would be good to note explicitly that meteorological drivers are correlated, it is not possible to fully separate them. They are driven by circulation patterns. This is analyzed in the SI but could be taken into account more in the main text at several instances. It should also be mentioned here that the choice of global circulation model driving WRF has an impact on the results, for the same RCP scenario, a different GCM may give the same global temperature change but different seasonal /regional impacts.

P3 l24 Since there may be nonlinear effects, the simulation with all drivers changed at once will not be the sum of the individual scenarios with one driver changed. This notion could be added to the text.

P6 l 85 Reference to EEA is not precise enough: which document/web page?

P9 I do not understand the approach towards rain episode: unit in Fig 2 is number of

hours (per year), but the number of episodes is mentioned in the text. I would call it total duration of rain if I understand your definition correctly (l 57-59). Is the threshold set to exclude drizzle? Number of rain events is something else.

P10 l68 Is there a main message from these correlations that should be mentioned here? Maybe conclusions that are used in the interpretation of the PM10 relationships to metereorological drivers?

P13 This section is difficult to follow. First relative changes due to climate change for EUR are discussed, then absolute concentrations for the Mediterranean and EUR are presented. For MED the impact of climate change is not presented, whereas this is the main subject of the paper. Transition to p14 is a surprise. Figure 8 and 9 are just mentioned without further explanation here. Why do you show them?

P14 Section 3.4: this section is quite long and repeats many well-known relations between PM10 and meteorology. Could you highlight the interesting parts (difference in distribution for sulfate) by reducing the description of the non-surprising parts? (e.g. nitrate analysis). I miss a short description of the behavior for the primary components PPM/POA/BC to start with.

P16 l36: but PBL and wind speed are highly correlated themselves, so this is no surprise.

P17 l59 I would say increase in concentration, not production, as you analyze concentration which is a results of production/chemistry, transport and deposition. How confident are you in the SOA scheme/literature, given the evolution in SOA parameterizations over the past 10 years?

P20 l90-96 think that changes in wet deposition are far more relevant than changes in RH and T for mineral dust, as it is hardly takes up water, and dust is dominated by inflow through the boundary of the model domain.

P20 l 5-9 Precipitation is difficult to model accurately, also for present-day conditions.

The analysis does not provide an indication of rain intensity/frequency, only total number of hours and total amount, as far as I understood

P21 not only accumulated effects but also taking into account nonlinearities/compensating effect.

P25 l2 Focus: EUR and MED get more or less same attention in the paper.

P27 l 30 But also secondary aerosols are relatively sensitive to advection, as it takes time to form them, but there are more processes involved.

P28 For sea salt emissions, I expect that changes in wind speed/circulation patterns and wet deposition will dominate over changes in salinity and density. Given the still large uncertainties in sea spray emission parameterizations, I would consider the relation between temperature, sea level and salinity that is mentioned irrelevant here and leave it out. There is a large difference between sea spray emissions in the Mediterranean and in the Baltic sea due to the far lower salinity of the latter, but I would by no means expect such a drastic change in salinity for the Mediterranean. Eventually sea level rise could give an effect on total area covered by the sea. Other land use changes are also not addressed in the paper, and they may have more impact (BVOC emissions, deposition) without being mentioned in the conclusions.

Minor textual comments

P1 14 dependency on temperature/humidity

P1 22 large number of components, with different origins and different behavior with respect to . . .

P2 28-29 formulate more precisely

P2 30 positive/negative: use enhance or reduce

P2 52 possible future changes

P3 20 composition (instead of content)

P6 l70: Land use does not change in the simulations (and leave out the last sentence)

P12 section numbers 3.4 and 3.1

P13 Figure 7 shows the PM10 concentrations and concentration changes for all different scenarios and subdomains, as well as the contributions of all the different PM10 components.

P13l 25 An interesting results is that sulfate concentrations show an increase...

P17 56 Our simulations are consistent with these results

P21 Typo in caption fig 10, 3rd line.

P22 l 51 climate effects were a few percent. I would not call them negligible, they are small but still visible

P24 l 77 COV not explained, this is first instance

P24 l 80. To be presented later: next paper or next section?

P27 l 32 In contrast (instead of on the contrary)

P27 l 34 emission changes show larger effects on non-dust and non-sea salt PM10 and PM2.5 components than changes in boundary conditions and climate conditions

P27 l 37 further investigation

P27 l 6 is dominated by dust

───────────────────────

---

## Referee Comment (RC2) · Anonymous Referee #3 · 25 Jan 2019

**Review of Future climatic drivers and their effect on PM 10 component in Europe and the Mediterranean Sea by Cholakian et al.**

This study proposes an in depth analysis of the potential evolution of PM and related drivers over Europe and part of the Mediterranean basin. To do so, the CHIMERE-WRF regional climate chemistry system is driven dynamically and chemically by the LMDZ INCA GCM following different RCPs climate and emissions scenarios. Anthropogenic Emission within the regional domain follow ECLIPSE scenarios. The effect of regional climate change, boundary conditions change and emission change are assessed using ad-hoc sensitivity tests.

Overall this study presents a lot of interesting information and deserves to be published in ACP. There are however a few points and comments to be addressed before that.

**Section 2 :**

Modelling framework :

For any regional climate study it is good to have information on the driving system, especially because both dynamic and chemistry are driven by LMDZ-INCA here. If possible provide references where LMDZ-INCA scenarios are analyzed in term of climate change ( e.g. CMIP5 intercomparison) and future PM conditions.

Are the LMDZ-INCA runs also driven by ECLIPSE emission for the chemistry part ?

Please specify the frequency of the chemical boundary coupling with CHIMERE. Is it monthly or higher frequency ? this could be relevant especially for dust outbreak simulations via the southern and eastern boundaries.

Vertical grid. The top of the model is 500 mb, but we know that Mediterranean basin could be influenced by long rage transport in the upper troposphere. Is there also a chemical boundary conditions at the top of the model, driven by INCA ?

Experimental design:

You choose to include natural emission change as part of regional climate change analysis. That makes senses but it should be clearly stated ( perhaps natural emission should not go under the air pollutant umbrella). It is clear that MEGAN is used for BVOC, apparently but do you have also sea-salt and dust production within your regional WRF/CHIMERE domain ?

The description of chemical BC experiment is a bit unclear to me. You mentioned that you considered two emission scenarios RCP and ECLIPSE for the global CTM. This does not reflect in table 1 however.

Also another question is what would have been the dust boundary conditions change provided by INCA if you had considered another climate scenario ? In the global forcing fields, are there a lot of differences between dust change simulated under different RCP projections ?

In general, PM boundary condition change is driven by climate ( and emission) change in the GCM. Caution should be taken in the final interpretation of BC change vs regional climate change, especially when discussing dust and the MED region.

**Section 3 : climate impact.**

Section 3.2 L 70. Actually for MED/RCP26 you have a slight increase of PM on figure 4 when in the text you mention -1.77% ?

L90-90.  Can the summer increase in all scenario be related to biogenic emission and if yes make the link with section 3.4.2 .

*Section 3.3 and 3.4*

Decrease in nitrate: just a side question , is there a significant trend in gas phase HNO3 ?

Sulfate : you mention the importance of aqueous formation can you confirm that just by looking at cloud cover trend given by wrf ?

BVOC : that could also explain the seasonal pattern of total PM change ( general decrease but increase in summer).

DUST :   Again, caution should be taken since regional climate change impact on dust sources, strongly determined by the Mediterranean due to southern boundary location.

**Section 4 .1**

Line 20.   See the above discussions. Dust boundary conditions change is related to Mediterranean climate change, as simulated by the GCM.

You mentioned land use change. Does the LMDZ-INCA simulation include CMIP land use change ?

Decrease of Sulfate : is it mainly related to a decrease in SO2 emission just outside the domain (northern Africa sources) that you could see from  the ECLIPSE scenario used to drive LMDZ-INCA ?

**Section 4.2**

The decrease in BSOA associated to a decrease in anthropogenic aerosol is indeed very interesting… but the magnitude of this decrease is quite "impressive" especially when compared to the impact  of biogenic emissions in a changing climate. How much confidence do we have in this result ?  Do you see a large decrease of oxidant activity in the chimere outputs ? Do you see a relative increase of isoprene and monoterpenes concentration?

**Section 5.**

In line with previous comment, the discussion between regional climate change and boundary condition effect should mention that Mediterranean climate change and dust activity are linked but could not be captured in a fully consistent way because of the choice of the CORDEX domain.

---

## Author Comment (AC1) · 26 Feb 2019

The authors thank both referees for their thorough and pertinent reviews, which certainly allow us to improve the paper. Below we provide answers to points raised by each of the referees.

In this response, bold parts in quotes are direct extracts of referee comments, blue italic parts are changes made in the article and black normal texts are answers/explanations on each comment made by the referees.

Referee 1 comments:

**Referee 1 general comments:**

> **"The paper presents a wealth of data on future climate, based on several climate simulations, emission scenarios, and boundary condition scenarios, for a domain covering Europe and the Mediterranean Sea. The data are analyzed in detail, with attention to all contributors to PM10 (PM2.5) and the impact of the different drivers. The analysis is done for the European continent, for the Mediterranean basin and two smaller subdomains covering parts of the Mediterranean area. The problem with the paper is that there is nearly too much information, and that it is difficult to remain focused on the main results. A major point of criticism is that the Mediterranean domain, and more so for the subdomains, are far smaller than the European domain. It is not completely fair to contrast them, and within Europe, different regimes could be found as well (e.g.EUROCORDEX regions). Also, the Mediterranean are is not even fully covered by the model and significantly affected by the boundary conditions. Nevertheless, the approach as such is not wrong, if one is clear on the limitations, and I would encourage the authors to motivate the choices more clearly."**

As a general answer to the comments made by the referee, the goal of this article is to explore the effects of climate change on the Mediterranean region, while exploring the European continent as a whole as well in order to provide a reference point for the study of the Mediterranean area. The simulations in this work were performed and analyzed in the context of the EUROCORDEX project in Colette et al (2013) for all the EUROCORDEX regions, Therefore any regimes existing in the EURPCORDEX sub-domains have been explored in the aforementioned study. In this work, we chose to provide a more detailed analysis for the European region compared to the aforementioned reference (in particular in relation with aerosol composition) regarding to PM concentrations as well as including the Mediterranean region which was not explored before. In order to study the Mediterranean region, a point of reference was needed, which was chosen to be the European continent as a whole. This choice makes more sense when taking into account the transportation of atmospheric components from the European region towards the Mediterranean area (apart from dust emissions). Also, the analysis pertaining the Mediterranean region answers directly to one of the goals of this campaign, studying the future conditions of this basin in different climatic conditions. About the Mediterranean sub-domain being at the southern borders of the domain, the following phrases were added in order to highlight the point raised by the referee.

> *It is important to keep in mind that, because of the location of the EUROCORDEX domain and the fact that the southern part of the Mediterranean is at the southern borders of the domain, the model might not be able to capture the effects of global climate change and dust activity in a fully consistent way, although the results show an important increase in dust concentrations because of long range transport. (p19, l1-5)*

**Referee 1 major comments:**

1. **"Abstract: No discrimination was made in the conclusion for the European area and for the Mediterranean area, whereas a lot of attention was given to the differences in the main text. Add a few sentences on the major differences. Dust is not an anthropogenic emission, so only long-range transport and boundary condition effects can be studied in the present set-up."**

We agree with this comment, the following modifications were made to the results part of the abstract:

*The results show that regional climate change causes a decrease in $PM_{10}$ concentrations in our scenarios (in both European and Mediterranean sub-domains), as a result of a decrease in nitrate, sulfate, ammonium and dust atmospheric concentrations in most scenarios. On the contrary, BSOA shows an important increase in all scenarios, showing more pronounced concentrations for the European sub-domain compared to the Mediterranean region. Regarding to the relationship of different meteorological parameters with concentrations of different species, nitrate and BSOA show strong temperature dependence, while sulfate is most strongly correlated with relative humidity. The temperature-dependent behavior of BSOA changes when looking at the Mediterranean sub-domain, showing more dependence to wind speed, because of the transported nature of BSOA existing in this sub-domain. A cumulative look at all drivers shows that anthropogenic emission changes overshadow changes caused by climate and long-range transport for both explored sub-domains, with the exception of dust particles for which long-range transport changes are more influential, especially in the Mediterranean basin. For certain species (such as sulfates and BSOA), for most explored sub-domains, the changes caused by anthropogenic emissions are to a certain extent reduced by boundary condition and regional climate changes. (p1-l26-40)*

2. **"P2 38-51: Here the motivation of the choice for Europe and Mediterranean is made, but could be improved. North-eastern Europe and Mediterranean are mentioned as hotspot, but in the next sentence Europe as a whole is mentioned. The relation to Charmex is only mentioned, not made in the paper. Should this come back in the conclusions?"**

Yes, North-Eastern Europe and the Mediterranean are in fact the two regions showing the highest sensitivity to climate change (according to the provided reference), while the Western Europe is among the second most sensitive areas. As an average, if we were to consider Europe as a whole, the sensitivity is still one of the highest in the world, as it is explained in the text as well. Some modifications were made in the text (added below) in order to better highlight this fact. The authors agree that the relationship of the article to the ChArmEx project can be better presented in the abstract, the introduction, as well as in the conclusions (added below).

*Two major sub-domains are explored, the European region and the Mediterranean basin, both areas showing high sensitivity to climate change. The Mediterranean area is explored in the context of the ChArMEx project, which examines the current and future meteorological and chemical conditions of the Mediterranean area. (p1 –l13-13)*

*Using the differences between historic and future precipitation and temperature for different regions and seasons in an ensemble of scenarios and models, he has shown that the Mediterranean and the north-eastern European area are more sensitive to climate change than the other regions of the world, followed by the Western Europe. According to his calculations, the European area (both western and eastern regions as an average) and the Mediterranean as a whole, are one of the most important hotspots for climate change. (p2, l25-30)*

*This work focuses on the Mediterranean area as well as the European area, since there have not been many studies focusing on the climate change drivers in the Mediterranean area, although this region might be highly sensitive to climate change, therefore directly responding to one of the major goals of the ChArMEx project, in the context of which the study was performed. (p18, l3-6)*

3. **"P3 It would be good to note explicitly that meteorological drivers are correlated, it is not possible to fully separate them. They are driven by circulation patterns. This is analyzed in the SI but could be taken into account more in the main text at several instances. It should also be mentioned here that the choice of global circulation model driving WRF has an impact on the results, for the same RCP scenario, a different GCM may give the same global temperature change but different seasonal /regional impacts."**

On the first point made by the referee, yes, it is true, as mentioned by the referee it has been mentioned in the article and treated in the SI. A phrase was added to the text of the article (added below) in order to mention this fact more explicitly.

*These meteorological parameters have interactions between themselves, showing correlations with each other since they are driven by circulation patterns. The values of correlations between different meteorological parameters examined in this work are shown in SI (SI4). (p8-l30-33)*

For the second point (a very good point as well), the following phrase was added to the same paragraph.

*Also it should be noted that if the GCM used to provide boundary conditions to the regional climate model was changed, the results seen here might be different (Olesen et al., 2007; Teichmann et al., 2013; Kerkhoff et al., 2015; Lacressonnière et al., 2016). (p8, l36-38)*

4. **"P3 l24 Since there may be nonlinear effects, the simulation with all drivers changed at once will not be the sum of the individual scenarios with one driver changed. This notion could be added to the text."**

The following phrase has been added to the introduction in order to highlight this comment.

*Finally, the simulations are compared to a series of simulations, where all the aforementioned drivers change at the same time, which can show us the overall impact of all which may be different to the sum of impacts of individual drivers due to non-linear effects. (p4, l1-3)*

5. **"P6 l 85 Reference to EEA is not precise enough: which document/web page?"**

The following URL was added as a footnote.

*https://www.eea.europa.eu/data-and-maps/indicators/global-and-european-temperature-8/assessment (p6, footnote)*

6. **"P9 I do not understand the approach towards rain episode: unit in Fig 2 is number of hours (per year), but the number of episodes is mentioned in the text. I would call it total duration of rain if I understand your definition correctly (l 57-59). Is the threshold set to exclude drizzle? Number of rain events is something else."**

Indeed, the limit is used to remove drizzle and the hours when the amount of rain was negligible. If the threshold was not used (and only a zero/non-zero treatment was done) almost all the hourly time steps

would be counted as having rain. The numbers provided in the images and discussed in the article correspond to the "number of rainy hours with an amount higher than the fixed threshold of the 25th percentile of all the simulations and all the domains". The number of different rain events has not been calculated, only the number of hours in which rain has occurred in each cell have been counted. The phrase "number of rain episodes" was changed to "total duration of rain" in all this section.

7. **"P10 l68 Is there a main message from these correlations that should be mentioned here? Maybe conclusions that are used in the interpretation of the PM10 relationships to meteorological drivers?"**

The following phrase was added to the text using the correlations represented in the SI (SI4).

> *The main points that should be taken into account about cross-correlations between meteorological parameters are the positive correlations between wind speed and boundary layer height, between wind speed and precipitation and also the anti-correlation of wind speed with relative humidity. All these correlations are above 0.6 as an absolute value. (p8, l32-35)*

8. **"P13 This section is difficult to follow. First relative changes due to climate change for EUR are discussed, then absolute concentrations for the Mediterranean and EUR are presented. For MED the impact of climate change is not presented, whereas this is the main subject of the paper. Transition to p14 is a surprise. Figure 8 and 9 are just mentioned without further explanation here. Why do you show them?"**

The section first describes the relative changes in the European sub-domain, then the differences between the Mediterranean and the European sub-domains are explored in form of absolute values. Figures 8 and 9 are used (and explored) later in the articles, they are just introduced in this section. The authors agree that this introduction should take place in the next section, therefore the paragraph removed from this section and the introduction of these two figures added to the next section (added below). The authors also agree that some information regarding the impacts of regional climate on the Mediterranean are necessary to be added to this section (added below).

> *Figure 8 shows the seasonal changes for nitrates, sulfates, BSOA and dust particles for all sub-domains. 2D concentration fields for nitrates, BSOA, sulfates and dust particles are shown in figure 9, for the season when each component shows its highest concentration (figure 8). (p11, l5-7)*

> *The relative changes in the Mediterranean domain compared to historic simulations are close to those for the European sub-domain for most species, albeit showing different intensities for most components. An interesting behavior is seen for BSOA, where the increase in the concentration of this component becomes more homogenous between the three future climatic scenarios in the Mediterranean basin with respect to continental Europe. The reason for this behavior is the origin of BSOA by advection over the Mediterranean sub-domain. Also, sulfates, while showing the same general behavior as in the European sub-domain, show lower changes between future and historic simulations over the Mediterranean Basin, resulting in a decrease in concentration in the RCP8.5 scenario. (p10, l31-37)*

9. **"P14 Section 3.4: this section is quite long and repeats many well-known relations between PM10 and meteorology. Could you highlight the interesting parts (difference in distribution for sulfate) by reducing the description of the non-surprising parts? (e.g. nitrate analysis). I miss a short description of the behavior for the primary components PPM/POA/BC to start with."**

The goal of this part was to explore every $PM_{10}$ component in the model that has apparent relationship with meteorological components or has a high contribution to the $PM_{10}$ concentration in any of the sub-domains, therefore the authors would like to leave all the information provided in this section the way it is. However, we agree with the referee that because of the length of the section and the amount of the information provided in it, it can be hard to see the principal points of the discussion. In order to address this issue, an effort was made to include what we find interesting and what we want to pass along in the conclusions part of the article. The authors also recognize primary species not appearing in this section. However, since their concentrations do not show strong changes with climate change alone or do not constitute a major component of the $PM_{10}$, they are discussed directly in section 4.2 (anthropogenic emission changes). However, following the referees suggestion, we added some specific sentences about the climate impact on primary particulate matter.

*Another interesting point from this study is that primary aerosol species such as primary organic aerosols (POA), anthropogenic secondary organic aerosols (ASOA), PPM and black carbon (BC) change only slightly under a future climate (when emissions are kept constant), both over Europe and the Mediterranean basin (a maximum of ±5% change for most of them). It should be noted here that this article deals with annual averages and not extreme events, regional climate changes can have strong effects on primary pollutant peaks as it is shown in Vautard et al. (2018). The evolution of these species will be again discussed in section 4.2 which respect to anthropogenic emission changes are discussed. (p9, l32-38)*

10. **"P16 l36: but PBL and wind speed are highly correlated themselves, so this is no surprise."**

That's true, the phrase was changed to:

*Another parameter that shows a strong anti-correlation with sulfate concentrations is the wind speed in spring and winter periods, which can be explained by the correlation between PBL height and wind speed (figure 10). (p12, l15-17)*

11. **"P17 l59 I would say increase in concentration, not production, as you analyze concentration which is a results of production/chemistry, transport and deposition. How confident are you in the SOA scheme/literature, given the evolution in SOA parameterizations over the past 10 years?"**

Yes, the word "production" was changed to the word "concentration" in order to be more precise.

The referee raises a very good point about the recent evolution of SOA simulation schemes, but which goes beyond the present study. In this work, we have used the standard aerosol scheme in CHIMERE (Bessagnet et al. 2009), in which representative hydrophilic and hydrophobic semi-volatile organic species are formed from various VOC precursors. In future work, we address the sensitivity of BSOA and SOA concentration changes with respect to several schemes, for a selected period of time, as we do not dispose of computer time for such a sensitivity analysis for the full 100 year period addressed here. Although the concentration change simulated by the three tested schemes (the CHIMERE standard scheme and two VBS schemes) are different , they all show the same pattern of climate response as observed in the current work, however with different intensities.

12. **"P20 l90-96 think that changes in wet deposition are far more relevant than changes in RH and T for mineral dust, as it is hardly takes up water, and dust is dominated by inflow through the boundary of the model domain."**

The authors agree with the referee about the wet deposition having a more important impact than RH and T. In order to show this better, we have added the following phrase:

*The impact of changing precipitation frequency/amount is the dominant factor in our simulations when it comes to dust concentration changes, since changes in precipitation patterns result in changes the wet deposition amounts. (p14, l10-12)*

**13. "P20 l 5-9 Precipitation is difficult to model accurately, also for present-day conditions. The analysis does not provide an indication of rain intensity/frequency, only total number of hours and total amount, as far as I understood."**

The referee is correct. In this study the total number of rainy hours in each year has been counted, the word duration used in this paragraph refers to that amount and not the duration of different rain episodes. The paragraph has been changed (added below) in order to highlight the point made by the referee.

*Finally, precipitation has been pointed out as a crucial, but difficult to model/predict parameter in both current and future meteorological/climatic simulations (Dale et al., 2001). In our study, the correlation of PM or PM components with the annual precipitation amount are generally weak (and sometimes even positive correlations are seen instead of expected anti-correlations). It has also been discussed, that total annual precipitation duration could be more impactful on the PM than the total amount (Dale et al., 2001). No correlation study has been performed with this parameter in this work, but the decreasing total precipitation duration in all scenarios could induce some increase in PM and its components. (p14, l26-32)*

**14. "P21 not only accumulated effects but also taking into account nonlinearities/compensating effect."**

The following phrase was added to highlight the point raised by the referee.

*Keep in mind that the series of simulations where all drivers change at the same time show not only the accumulative impacts of all drivers, but also the non-linear relationships that exists between different drivers. (p15, l8-10)*

**15. "P25 l2 Focus: EUR and MED get more or less same attention in the paper."**

The referee is correct, the phrase was changed to:

*This work focuses on the Mediterranean area as well as the European area, since there have not been many studies focusing on the climate change drivers in the Mediterranean area, although this region might be highly sensitive to climate change, therefore directly responding to one of the major goals of the ChArMEx project, in the context of which the study was performed. (p18, l3-6)*

**16. "P27 l 30 But also secondary aerosols are relatively sensitive to advection, as it takes time to form them, but there are more processes involved."**

The sensitivity of BSOA concentrations to advection is discussed in section 3.4.2, where its relationship with wind speed for the Mediterranean area has been pointed out and the following sentences were added to clarify more (added below). The difference in the sensitivity of dust and BSOA to advection lies in the fact that the emissions of dust is more episodic, while that of BSOA is a constant source especially during summer. Also, the advection of the dust emissions happens from outside of our domain, while the transportation of BSOA happens inside the domain. It should also be taken into account that averaged seasonal meteorological parameters and concentrations are compared for each sub-domain, therefore

what is shown here does not correspond to local correlations, but regional ones. These reasons explain why the model can predict the advection of BSOA to the Mediterranean area, while it seems to miss that of dust aerosols. The phrase below has been modified in order to present this difference.

*The correlation of BSOA concentration in the Mediterranean with wind speed corresponds to the point raised above about the advective nature of BSOA concentrations in this sub-domain, while the high correlations of PBL height and RH with BSOA come from the correlation of wind speed with these two parameters. (p13, l17-20)*

*… while dust concentrations present a weak relationship with the tested meteorological parameters since their changes are related to advection from outside the model domain and therefore not captured by local correlation analysis. (p18, l35-37)*

17. **"P28 For sea salt emissions, I expect that changes in wind speed/circulation patterns and wet deposition will dominate over changes in salinity and density. Given the still large uncertainties in sea spray emission parameterizations, I would consider the relation between temperature, sea level and salinity that is mentioned irrelevant here and leave it out. There is a large difference between sea spray emissions in the Mediterranean and in the Baltic Sea due to the far lower salinity of the latter, but I would by no means expect such a drastic change in salinity for the Mediterranean. Eventually sea level rise could give an effect on total area covered by the sea. Other land use changes are also not addressed in the paper, and they may have more impact (BVOC emissions, deposition) without being mentioned in the conclusions."**

The part corresponding to sea salt changes was removed taking into account the referees' remarks. The following phrases regarding land-use changes were added.

*Land-use changes, apart from effects on dust emission mentioned above, can affect the emissions of BVOC, which can change the future concentrations of BSOA, they also can change the deposition of different species in future scenarios. (p20, l2-4)*

**Referee 1 minor textual comments:**

1. **P1 14 dependency on temperature/humidity**

The section was changed at the request of the referee, therefore the comment has been modified before.

2. **P1 22 large number of components, with different origins and different behavior with respect to …**

Changed to:

*PM comprise a large number of components, with different origins and diverse behaviors with respect to meteorological parameters. (p2, l5-6)*

3. **P2 28-29 formulate more precisely**

Changed to:

*Thus, when exploring future air quality, it is important to take into account that different drivers can have different impacts while having non-linear interactions among themselves. Therefore it is interesting to explore the effects of each driver separately. (p2, l2-14)*

4. **P2 30 positive/negative: use enhance or reduce**

Changed.

**5. P2 52 possible future changes**

Changed.

**6. P3 20 composition (instead of content)**

Changed.

**7. P6 l70: Land use does not change in the simulations (and leave out the last sentence)**

Sentence was changed to:

*The same unchanged land-use data from Globcover (Arino et al., 2008) with a base resolution of 300 ×300m² have been used in different series of simulations. (p5, l21-22)*

**8. P12 section numbers 3.4 and 3.1**

Changed.

**9. P13 Figure 7 shows the PM10 concentrations and concentration changes for all different scenarios and subdomains, as well as the contributions of all the different PM10 components.**

Changed.

**10. P13l 25 An interesting results is that sulfate concentrations show an increase…**

Fact changed to result in that sentence.

**11. P17 56 Our simulations are consistent with these results**

Changed.

**12. P21 Typo in caption fig 10, 3rd line.**

Corrected.

**13. P22 l 51 climate effects were a few percent. I would not call them negligible, they are small but still visible**

The word "negligible" was changed to "smaller".

**14. P24 l 77 COV not explained, this is first instance**

The definition of VOC as well as BVOC were added to section 3 and section 2.4, when they were first used.

**15. P24 l 80. To be presented later: next paper or next section?**

Sentence changed to:

*(presented in section 4.3)*

**16. P27 l 32 In contrast (instead of on the contrary)**

Changed.

**17. P27 l 34 emission changes show larger effects on non-dust and non-sea salt PM10 and PM2.5 components than changes in boundary conditions and climate conditions**

Changed to:

*Emission changes show the largest effect on all non-sea salt and non-dust $PM_{10}/PM_{2.5}$ components. (p19, l6)*

**18. P27 l 37 further investigation**

Changed**.**

**19. P27 l 6 is dominated by dust**

Changed.

Referee 2 comments:

**"For any regional climate study it is good to have information on the driving system, especially because both dynamic and chemistry are driven by LMDZ-INCA here. If possible provide references where LMDZ-INCA scenarios are analyzed in term of climate change (e.g. CMIP5 intercomparison) and future PM conditions."**

The LMDZ-INCA runs used in this work have been analyzed in Szopa et al (2013). Intercomparison studies regarding these runs are analyzed in Shindell et al (2013) and Young et al (2013) in the framework of the ACCMIP project. The following passage has been added to the article to explain this more.

*The LMDZ-INCA runs used in this study have been analyzed in Szopa et al. (2013) and Markakis et al. (2014) and inter-comparisons of the same runs with other global chemistry-transport models have been analyzed in Shindell et al (2013) and Young et al (2013) in the framework of the ACCMIP experiment. (p5, l4-7)*

**"Are the LMDZ-INCA runs also driven by ECLIPSE emission for the chemistry part?"**

Two runs of LMDZ-INCA are used in this study, one for all the simulation series (with the exception of the following two series), the second one for the series of simulations used to explore the boundary condition change impacts and the series of simulations in which all drivers change at the same time. The first series of simulations uses RCP emissions, while the second one uses ECLIPSE emissions. Table one was modified in order to show the emissions used for each series. More information about the LMDZ-INCA runs are provided in Szopa et al, (2013), Shindell et al (2013), Young et al (2013) and Markakis et al (2014).

**"Please specify the frequency of the chemical boundary coupling with CHIMERE. Is it monthly or higher frequency? This could be relevant especially for dust outbreak simulations via the southern and eastern boundaries."**

Monthly data is used for the boundary condition input fields for all the series of simulations discussed in this study. Given that background changes over long periods of time are discussed in this study, monthly inputs provide the necessary information, and it was not possible to store hourly large scale model output. This induces an unavoidable inconsistency between three-hour meteorology and the monthly dust fields. The following phrase has been added in order to highlight this point.

*Monthly climatological fields are used as the boundary condition inputs; since background changes over long periods of time and it was not possible to store hourly global model output to create hourly varying boundary conditions. This induces an unavoidable inconsistency between meteorology and dust fields. (p5, l7-10)*

**"Vertical grid. The top of the model is 500 mb, but we know that Mediterranean basin could be influenced by long rage transport in the upper troposphere. Is there also a chemical boundary conditions at the top of the model, driven by INCA?"**

Yes, boundary chemical conditions are used from the top of the model as well, from the same source and with the same frequency as for horizontal boundary conditions.

**"You choose to include natural emission change as part of regional climate change analysis. That makes senses but it should be clearly stated (perhaps natural emission should not go under the air pollutant umbrella). It is clear that MEGAN is used for BVOC, apparently but do you have also seasalt and dust production within your regional WRF/CHIMERE domain?"**

Yes, both sea salt and dust emissions are taken into account inside the simulation domain and these emissions are modified under a changing climate. This becomes clear in the following sentences in the manuscript.

*Dust emissions are taken into account inside the simulation domain based on the method proposed by (Marticorena and Bergametti, 1995).(p5,l23-24)*

Sea salt emissions are treated by the Monahan, 1986 method in the model.

**"The description of chemical BC experiment is a bit unclear to me. You mentioned that you considered two emission scenarios RCP and ECLIPSE for the global CTM. This does not reflect in table 1 however."**

As mentioned for the previous comment made by the referee, two runs of LMDZ-INCA are used in this study, one for all the simulation series (with the exception of the following two series), the second one for the series of simulations used to explore the boundary condition change impacts and the series of simulations in which all drivers change at the same time. The first series of simulations uses RCP emissions, while the second one uses ECLIPSE emissions. Table one was modified in order to show the emissions used for each series. More information about the LMDZ-INCA runs are provided in Szopa et al, (2013), Shindell et al (2013), Young et al (2013) and Markakis et al (2014).

**"Also another question is what would have been the dust boundary conditions change provided by INCA if you had considered another climate scenario? In the global forcing fields, are there a lot of differences between dust changes simulated under different RCP projections?"**

Szopa et al (2013) discusses this point when exploring the global runs that we use as boundary conditions. A general decrease in the aerosol burden is seen in all scenarios, while an increase in dust concentration is observed in all scenarios as well. This increase becomes more important for more severe scenarios (RCP8.5>RCP6>RCP4.5>RCP2.6). So, if instead of RCP4.5 RCP8.5 was used, it normally should have resulted in a higher increase in dust concentrations towards the end of the century according to the aforementioned reference. The reason for this increase has been associated to weakened wet deposition around 40°N in the global simulations.

**"In general, PM boundary condition change is driven by climate (and emission) change in the GCM. Caution should be taken in the final interpretation of BC change vs regional climate change, especially when discussing dust and the MED region."**

Yes, the boundary condition change corresponds to the global climate changes as well as global emission changes. The following phrase was added to section 4 in order to reflect this fact. It is affected by uncertainty, so as the impact of regional climate changes. However, comparison between the regional climate change and the BC impact are for all compounds sufficiently strong to conclude on a major driver, very probably beyond uncertainty. The following phrase was added to section 4 in order to reflect this fact.

> *It should be noted that boundary conditions are taken from a global chemistry-transport model, therefore, changing the boundary conditions corresponds to changing the global climate and the global anthropogenic emissions at the same time. However, comparison between the regional climate change and the BC impact are for all compounds sufficiently strong (see below) to conclude on a major driver, very probably beyond uncertainty. (p15, l2-6)*

**"Section 3.2 L 70. Actually for MED/RCP26 you have a slight increase of PM on figure 4 when in the text you mention -1.77%?"**

Yes, the value in the text has been corrected to 0.9%.

**"L90-90. Can the summer increase in all scenario be related to biogenic emission and if yes make the link with section 3.4.2."**

The summer increase is indeed partly due to the biogenic SOA formation, a link is added to section 3.4.2 in this section.

> *This increase for the summer period is due to BVOC emission increases, which will be discussed in section 3.4.2. (p9, l18-19)*

**"Decrease in nitrate: just a side question, is there a significant trend in gas phase HNO3?"**

Yes, there is a significant positive trends for all scenarios for gas-phase $HNO_3$.

**"Sulfate: you mention the importance of aqueous formation can you confirm that just by looking at cloud cover trend given by wrf?"**

Yes, there is a correlation between cloud cover and the sulfate concentration, not a particularly strong one though (average of 0.42 of correlation for all scenarios and all seasons).

**"BVOC: that could also explain the seasonal pattern of total PM change (general decrease but increase in summer)."**

Yes, it does, a sentence has been added to this section (and to the section before per referee's previous comment) to address this point.

> *This increase in BSOA reflects the summer increase in the $PM_{10}$ mentioned in section 3.2 as well. (p13, l5-6)*

**"DUST: Again, caution should be taken since regional climate change impact on dust sources, strongly determined by the Mediterranean due to southern boundary location."**

See the response for the comment below.

**"Line 20. See the above discussions. Dust boundary conditions change is related to Mediterranean climate change, as simulated by the GCM."**

The following phrases were added to section 4 and the conclusion in order to reflect this issue.

*It should be noted that boundary conditions are taken from a global chemistry-transport model, therefore, changing the boundary conditions corresponds to changing the global climate and the global anthropogenic emissions at the same time. (p15, l2-6)*

*Also, the Mediterranean Sea is located at the southern borders of the domain used for this study, therefore, it should be noted that although the results of dust concentration changes seen in this study are consistent with the existing literature, the model might not be capable of consistently capturing the relationship between boundary condition changes and southern parts of the Mediterranean because of the placement of the domain. This is not the case for the European sub-domain. (p15, l36-p16, l2)*

*It is important to keep in mind that, because of the position of the EUROCORDEX domain and the fact that the southern part of the Mediterranean is at the southern borders of the domain, the model might not be able to capture the effects of global climate change and dust activity in a fully consistent way, although the results show an important increase in dust concentrations because of long range transport. (p19, l1-5)*

**"You mentioned land use change. Does the LMDZ-INCA simulation include CMIP land use change?"**

According to Szopa et al. (2013), land use changes have not been considered in the LMDZ-INCA model.

**"Decrease of Sulfate: is it mainly related to a decrease in SO2 emission just outside the domain (northern Africa sources) that you could see from the ECLIPSE scenario used to drive LMDZ-INCA?"**

The referee is correct, it is mainly due to the emission reductions just outside of the domain.

**"The decrease in BSOA associated to a decrease in anthropogenic aerosol is indeed very interesting… but the magnitude of this decrease is quite "impressive" especially when compared to the impact of biogenic emissions in a changing climate. How much confidence do we have in this result? Do you see a large decrease of oxidant activity in the chimere outputs? Do you see a relative increase of isoprene and monoterpenes concentration?"**

There are several factors that could cause the decrease that we see in our BSOA concentrations because of anthropogenic emission changes. In the article, decrease in seed aerosol for formation of new SOA and change in the equilibrium of SVOC because of changes in the anthropogenic VOC emissions have been mentioned. Another reason, as mentioned by the referee, could be a change in the oxidant levels because of changes in anthropogenic emissions. As to respond to the two questions asked by the referee, yes, there is a decrease in oxidant levels which is manifested by an increase in isoprene/mono-terpene concentrations (under constant emissions in this scenario). Thus, in our opinion, atmospheric chemistry could cause such a decrease in BSOA concentrations because of anthropogenic emissions changes. In line with this issue, same impact was seen in a study in preparation by Ciarelli et al (2019) with also important intensity, when looking at an ensemble of simulations in the framework of the EURODELTA multi-model experiment. Sartelet et al. (2012) see an important change in SOA concentration in their simulation as well when changing the anthropogenic emissions. So we are confident in the effect, as it

is seen by many models, but at this time, we cannot explain it quantitatively. The following phrases have been added to the article:

> *Same impact was seen in a study in preparation by Ciarelli et al. (2019) with also important intensity, when looking at an ensemble of simulations in the framework of the EURODELTA multi-model experiment. Sartelet et al. (2012) see an important change in SOA concentration in their simulation as well when changing the anthropogenic emissions. (p19,l9-13)*

> **"In line with previous comment, the discussion between regional climate change and boundary condition effect should mention that Mediterranean climate change and dust activity are linked but could not be captured in a fully consistent way because of the choice of the CORDEX domain."**

The following phrases were added to section 4 and the conclusion in order to reflect this issue (same modifications as another comment of the referee).

> *Also, the Mediterranean Sea is located at the southern borders of the domain used for this study, therefore, it should be noted that although the results of dust concentration changes seen in this study are consistent with the existing literature, the model might not be capable of consistently capturing the relationship between boundary condition changes and southern parts of the Mediterranean because of the placement of the domain. This is not the case for the European sub-domain. (p15, l36-p16, l2)*

> *It is important to keep in mind that, because of the position of the EUROCORDEX domain and the fact that the southern part of the Mediterranean is at the southern borders of the domain, the model might not be able to capture the effects of global climate change and dust activity in a fully consistent way, although the results show an important increase in dust concentrations because of long range transport. (p19, l1-5)*